# Test-time Adaptation in Non-stationary Environments via Adaptive Representation Alignment

**Zhen-Yu Zhang**
Center for Advanced Intelligence Project, RIKEN
zhen-yu.zhang@riken.jp

**Zhiyu Xie**
Stanford University
zhiyuxie@stanford.edu

**Huaxiu Yao**
UNC-Chapel Hill
huaxiu@cs.unc.edu

**Masashi Sugiyama**
Center for Advanced Intelligence Project, RIKEN
Graduate School of Frontier Sciences, The University of Tokyo
sugi@k.u-tokyo.ac.jp

## Abstract

Adapting to distribution shifts is a critical challenge in modern machine learning, especially as data in many real-world applications accumulate continuously in the form of streams. We investigate the problem of sequentially adapting a model to non-stationary environments, where the *data distribution is continuously shifting* and *only a small amount of unlabeled data are available each time*. Continual test-time adaptation methods have shown promising results by using reliable pseudo-labels, but they still fall short in exploring representation alignment with the source domain in non-stationary environments. In this paper, we propose to leverage non-stationary representation learning to adaptively align the unlabeled data stream, with its changing distributions, to the source data representation using a sketch of the source data. To alleviate the data scarcity in non-stationary representation learning, we propose a novel *adaptive representation alignment* algorithm called Ada-ReAlign. This approach employs a group of base learners to explore different lengths of the unlabeled data stream, which are adaptively combined by a meta learner to handle unknown and continuously evolving data distributions. The proposed method comes with nice theoretical guarantees under convexity assumptions. Experiments on both benchmark datasets and a real-world application validate the effectiveness and adaptability of our proposed algorithm.

## 1 Introduction

Machine learning algorithms, particularly deep learning models, have achieved remarkable success when the test data share the same distribution of the training data. However, in many real-world applications, the learning environment is changing over time, resulting in the test data inevitably encountering natural variations. For example, once an autonomous driving model is deployed, external factors such as weather changes (e.g., snow, frost, or fog) and internal factors like sensor degradation (e.g., causing Gaussian noise, defocus, or blur) can negatively impact its performance. Unfortunately, many well-trained models are highly sensitive to such distribution shifts and may suffer significant performance degradation, even when the shifts are minor [12]. Moreover, once deployed, models often lack access to the original training data, making it essential to equip the learning system with the ability to adapt to non-stationary environments in test time [29, 45].

A promising line of research is known as *test-time adaptation* (TTA), which focuses on adapting models to new environments using only unlabeled data. Pioneering approaches consider adaptation to a fixed distribution, including introducing auxiliary self-supervised learning tasks [30, 28] or employing

38th Conference on Neural Information Processing Systems (NeurIPS 2024).

entropy minimization to update the model [26, 32]. In addition, aligning the representation of unlabeled data back to the source representation has also been proposed to improve performance [28, 10]. More recently, several seminal works have explored *continual test-time adaptation* with unlabeled data streams, where data arrive continuously, and the distribution may change over time. These algorithms often rely on generating and selecting high-confidence pseudo-labels to update the model, with techniques such as reliable entropy minimization [34, 25]. Non-parametric approaches have also be proposed, leveraging both source labels and pseudo-labeled data for adaptation [40]. Despite these advances, the exploration of effective representation alignment with the source data representation in non-stationary environments remains underexplored.

In this paper, we propose a non-stationary representation learning approach to adaptively align the unlabeled data stream with changing distributions to the source data representation, leveraging the marginal information from the source distribution. Our framework follows a common configuration used in prior work, which assumes the model comprises a representation learning module and a linear classifier [30, 32]. The core idea is to continually update the representation learning module so that it projects non-stationary unlabeled data at each time step to a distribution aligned with the source data representation, relying solely on the source marginal distribution. Different from prior representation alignment methods for fixed distributions [23, 28], the key challenge in non-stationary representation learning lies in the limited amount of data available at each time. Updating the model with only a single batch of data can introduce high variance due to the small sample size, while relying on the entire data stream risks high bias as the distribution evolves over time.

To handle the challenges posed by the unknown and ever changing distributions and limited data in each round, we propose a novel algorithm named *adaptive representation alignment* (Ada-ReAlign). Drawing inspiration from recent advances in non-stationary online learning [38, 43, 41], the representation learning module employs a group of base models learning with varying window sizes to explore different part of the data stream, along with a meta learner that adaptively combines their output to learn a well-aligned representation. We then further refine the linear classifier by applying entropy minimization, with regularization to the initial one, based on the aligned the data representation. Our algorithm enjoys nice theoretical guarantees with convex models and loss functions, providing a solid foundation for its empirical success. Through benchmark experiments and a real-world application, we demonstrate the effectiveness and efficiency of our method in improving performance under non-stationary environments. We summarize our contributions as follow:

(1) We propose *non-stationary representation learning* for continual test-time adaptation, which adaptively aligns non-stationary data with the source representation using a sketch of source data.

(2) We propose a novel online learning algorithm that adaptively aligns representations to the source by exploring variations across different periods of the data. Theoretical analysis demonstrates that the proposed method approximates the optimal model sequence for convex losses and models.

(3) We demonstrate the effectiveness and efficiency of our method for test-time adaptation in non-stationary environments through benchmark experiments and a real-world application.

**Organization.** We first introduce related work in Section 2. In Section 3, we present our main results, including the proposed algorithm with corresponding theoretical analysis. Section 4 reports the experiment results, followed by the conclusion in Section 5. All proofs and omitted details are provided in the Appendix.

## 2 Related Work

In this section, we present the most relevant literature related to our learning setting and the techniques used in our approach. A more detailed description is provided in **Appendix B**.

**Continual Test-time Adaptation in Non-stationary Environments.** Recently, continual test-time adaptation in non-stationary environments has gained significant attention. Due to the evolving nature of streaming data, the data distribution naturally changes over time. A simple idea involved recovering the model weights from the initial model after each adaptation step of a mini-batch, such as MEMO [39], episodic Tent [32], or DDA [10]. However, these sequential one-step adaptation methods could be insufficient because they only explore the limited number of data at each round, and thus cannot explore the knowledge of the accumulated historical data.

To tackle this challenge, CoTTA [34] was introduced, generating robust pseudo-labels through a weighted average of historical models while preserving the initial model's parameters. It also stochastically replaces the model's parameters with the initial model's parameters at each round to resist distribution change. Building on this, EcoTTA [27] and BcoTTA [20] improved parameter and memory efficiency during continual adaptation. Similarly, SAR [25] updated the model based on reliable entropy and reset it to its initial state whenever the entropy exceeds a predefined threshold. AdaNPC [40] resisted non-stationarity by constructing a memory buffer to store historical distributions. Although these methods have demonstrated empirical success in various real-world applications, they rely on prior knowledge to estimate pseudo-labels or require access to source-labeled data during adaptation.

**Adapting to Non-stationary Environments with Offline Labeled Data.** This line of research focuses on specific types of distribution shifts, employing adaptive learning with weighted source-labeled data to handle continuous distribution changes. Previous studies have tackled the challenge of non-stationarity in the context of online label shift, where only the class priors change. In such scenarios, an unbiased loss estimator is used to estimate the loss at each round, enabling dynamic regret minimization in non-stationary environments [2]. Another work investigated the problem of continuous covariate shift, where only the input distribution changes [41]. They reframed this problem as an online density ratio estimation task and proposed a generic reduction of the density ratio estimation problem to dynamic regret optimization. However, these approaches assume access to the offline training data for adaptation, which may not always be feasible in practice.

# 3 Algorithm and Theory

We start by introducing the notations and problem formulation, followed by a detailed explanation of the proposed Ada-ReAlign algorithm and its theoretical analysis.

## 3.1 Problem Formulation

In this part, we first formulate the learning problem. We assume access to a well-trained model on source data, along with a sketch of them. During adaptation, unlabeled data arrive sequentially from non-stationary environments, where the underlying data distribution could change over time. Following previous work [30, 32], we assume the model consists of a representation learning module and a linear function as classification module. Let $\phi_t(\cdot) : \mathcal{X} \mapsto \mathbb{R}^d$ be the representation learning module, and let the linear function be represented by a matrix $\mathbf{w}_t \in \mathbb{R}^{k \times d}$ where $k = |\mathcal{Y}|$. Thus, the prediction model is defined as $f_t(\cdot) = \langle \mathbf{w}_t, \phi_t(\cdot) \rangle$. We denote the representation module of the well-trained initial model on the source data by $\phi_0(\cdot)$ and its corresponding linear function by $\mathbf{w}_0$.

As streaming data are collected in non-stationary environments, the underlying data distribution remains both *unknown and ever-changing*. The learning task is framed as a sequential prediction problem over $T$ rounds, with $T > 0$. At each round $t \in [T] := \{1, \ldots, T\}$, the learner receives a batch of unlabeled data $S_t = \{\mathbf{x}_{t,i}\}_{i=1}^{N_t}$, sampled independently and identically from the distribution $\mathcal{D}_t$, where $N_t \geq 1$ and $\mathcal{D}_t$ could change over time. Our objective is to learn a sequence of models $\{f_t\}_{t=1}^T$ that perform well across the evolving distributions $\{\mathcal{D}_t\}_{t=1}^T$.

We define $\mu_0 \in \mathbb{R}^d$ and $\Sigma_0 \in \mathbb{R}^{d \times d}$ as the mean and covariance of the representation distribution of the source data with the initial model, where $\mu_0 = \mathbb{E}_{\mathbf{x} \sim \mathcal{S}_0}[\phi_0(\mathbf{x})]$, $\Sigma_0 = \mathbb{E}_{\mathbf{x} \sim \mathcal{S}_0}[(\phi_0(\mathbf{x}) - \mu_0)^T(\phi_0(\mathbf{x}) - \mu_0)]$, and $d$ denotes the dimensionality of the feature embedding. This approach of sketching the source data does not rely on label information from the source data, which is particularly advantageous in tasks involving privacy concerns. Moreover, marginal information can be generated using coreset techniques [31, 16], eliminating the need for direct access to the original source data.

Following the online learning literature [11], we use dynamic regret as the performance measure. The performance of the model sequence $\{f_t\}_{t=1}^T$ is evaluated through the average excess risk, defined as:

$$\text{D-Regret}(\{f_t\}_{t=1}^T) := \sum_{t=1}^T R_t(f_t) - \sum_{t=1}^T R_t(f_t^*), \tag{1}$$

where $R_t(f) = \mathbb{E}_{(\mathbf{x},y) \sim \mathcal{D}_t}[\ell(f(\mathbf{x}), y)]$ is the expected loss at time $t$ with loss function $\ell$, $f_t^* \in \arg\min_{f \in \mathcal{F}} R_t(f)$ represents the corresponding optimal model in the hypothesis space at each round.

When the distribution of unlabeled data stream is fixed, e.g., $\mathcal{D}_t = \mathcal{D}_1$, $\forall t \in [T]$, this formulation recover back to the previous setting of TTA to a stationary environment [30, 32]

**Remark 1** (Dynamic Regret and Catastrophic Forgetting). The dynamic regret defined in Eqn. (1) quantifies the difference between the performance of the learned model and the optimal model at each time step. In our formulation, the data distribution at any given time step can correspond to any previously encountered distribution, and the model has no prior knowledge of this distribution during prediction. Therefore, if the performance of the model remains comparable to that of the optimal model that minimizes the expected loss at that time step, this indicates that the learned model successfully retains previously acquired knowledge and mitigates the catastrophic forgetting issue.

## 3.2 Representation Alignment with Source Sketch

To adapt the model to a new target domain, a natural intuition is that if the data in the target domain can be accurately projected back to their representation in the source domain, the well-trained source model can be reused for predictions. Based on this idea, we keep the classification module fixed by setting $\mathbf{w}_t = \mathbf{w}_0$ and adapt the representation learning module $\{\widehat{\phi}_t\}_{t=1}^T$ to ensure that the representation of the non-stationary unlabeled data stream aligns closely with that of the source data.

We employ dynamic regret as the performance measure for representation learning in non-stationary environments and define the objective as:

$$\min_{\phi_t} \sum_{t=1}^T L_t(\phi_t, \phi_0) - \sum_{t=1}^T L_t(\phi_t^*, \phi_0), \tag{2}$$

where $L_t(\cdot, \cdot)$ represents the representation discrepancy between $\phi_t$ and $\phi_0$ in each round, where $\phi_t$ is the model learned at round $t$. Here $\phi_t^*$ denotes the optimal representation learning function, defined as $\phi_t^* = \arg\min_\phi L_t(\phi, \phi_0)$.

We now define the loss function $L_t(\cdot, \cdot)$. Given the challenges in accurately estimating representation discrepancy, we propose to use a surrogate loss function to approximate it. Inspired by prior work on representation learning with deep neural networks [37, 28], we model the representation distribution using a Gaussian approximation. Specifically, we measure the discrepancy as the gap in mean and covariance between the projected unlabeled data and the source data representation, defined as follows:

$$L_t(\phi, \phi_0) = \|\mu_t - \mu_0\|_2^2 + \lambda \|\Sigma_t - \Sigma_0\|_F^2$$

where $\lambda$ is the hyperparameter, $\mu_t = \mathbb{E}_{\mathbf{x}\sim\mathcal{D}_t}[\phi_t(\mathbf{x}_t)]$, $\Sigma_t = \mathbb{E}_{\mathbf{x}\sim\mathcal{D}_t}[(\phi_t(\mathbf{x}) - \mu_t)^T(\phi_t(\mathbf{x}) - \mu_t)]$, $\|\cdot\|_2$ denotes the Euclidean norm, and $\|\cdot\|_F$ denotes the Frobenius norm. To improve numerical stability, we also add the identity matrix multiplied with a small constant to each covariance matrices $\Sigma_t$ to reduce their condition number.

Since we only have empirical data at each round, we use the empirical loss to approximate the discrepancy. We define $\widehat{\mu}_t = \sum_{i=1}^{n_t} \phi_t(\mathbf{x}_i)/n_t$ and $\Sigma_t = \sum_{i=1}^{n_t}[(\phi_t(\mathbf{x}) - \widehat{\mu}_t)^T(\phi_t(\mathbf{x}) - \widehat{\mu}_t)]/n_t$, thus we have the empirical estimation of the divergence,

$$\widehat{L}_t(\phi, \phi_0) = \|\widehat{\mu}_t - \widehat{\mu}_0\|_2^2 + \|\widehat{\Sigma}_t - \widehat{\Sigma}_0\|_F^2 \tag{3}$$

At first glance, it may seem straightforward to minimize the loss defined in Eqn. (3) to update the representation learning module. However, the limited amount of data available in each round poses a challenge to obtain a well-generalized model. Therefore, we leverage online learning techniques to reuse a suitable number of historical data, ensuring effective learning of the representation model.

**Remark 2** (Comparison with Distribution Alignment Approaches). Aligning the representations of unlabeled data with those of source data has been explored in works such as [23, 28]. These studies primarily focus on adapting the model to a fixed domain with a large amount of unlabeled data. Our problem involves adaptation to non-stationary environments, where the data distribution can change continuously with a limited number of data available in each round. This non-stationary setting requires the development of novel methods to adaptively leverage historical data for adaptation.

The proposed discrepancy measure aligns the global representation distribution between the source data and the new unlabeled data. Additionally, exploring class-specific prototypes can further enhance performance [17, 28], which we leave for future investigation.

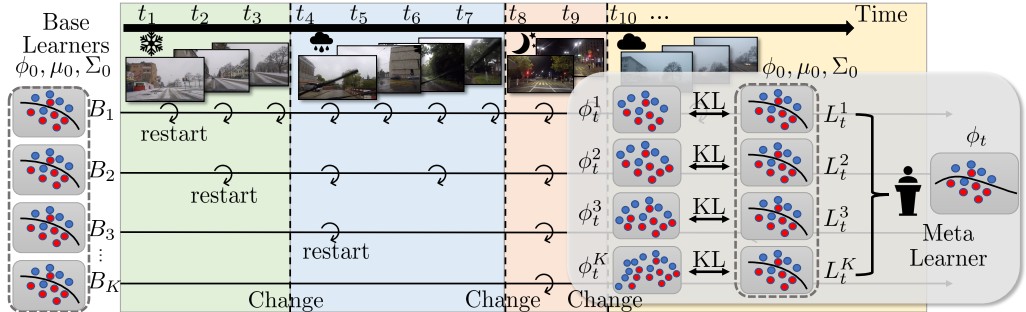

**Figure 1:** An illustration of our problem and the Ada-ReAlign algorithm. The test data accumulate over time with changing distributions, and only a limited number of unlabeled data are available at each step. Initially, an offline model and the statistics of the offline data are provided, followed by continuous adaptation to the new distributions. Ada-ReAlign is composed of a group of base learners and a meta learner. Each base learner operates with a different window size by restarting, learning representations for its respective period by minimizing the discrepancy from the source representation. The outputs from the base learners are then combined by the meta learner to produce the final representation.

### 3.3 Adaptive Representation Alignment

In this part, we present the proposed algorithm. Inspired from the *online ensemble* framework [43] developed in recent research on non-stationary online learning, we propose a two-layer adaptive learning algorithm. This approach is designed to handle the unknown change in data distribution and the limited data available in each round.

**Base Learner.** We construct a set of base learners $\{\phi^i\}_{i=1}^K$, each with a different learning window size. These base learners initialize their parameters as $\phi_0$. As shown in Figure 1, at each round, they perform online gradient descent using the loss defined in Eqn. (3), i.e,

$$\phi_{t+1}^i = \phi_t^i - \nabla_\phi \widehat{L}_t(\phi_t^i, \phi_0). \tag{4}$$

In addition to performing gradient descent, each base learner is assigned a learning window of varying size, determined by its index. As an example, the $i$-th base learner trains on data segments of length $2^i$. At each round $t = 2^i$, its representation function $\phi_t^i$ is re-initialized to the initial state $\phi_0$ and a new learning process is initiated using online gradient descent – a procedure we refer to as a "restart".

Intuitively, when the data distribution shifts gradually or stabilizes in a new environment, the base learner that leverages the entire historical dataset tends to perform well. In contrast, when the distribution undergoes abrupt changes, a base learner that frequently restarts and relies only on recent data can achieve competitive performance [18]. The flexibility of exploring a set of base learners allows us to design a meta learner that strategically combines these base learners, optimizing the overall performance of the ensemble algorithm.

**Meta Learner.** As shown in Figure 1, we employ a meta learner to combine the base learners that learn on different time length. To implement the meta learner, we employ the AdaNormalHedge algorithm with the geometric covering scheme [24]. At each round, the meta learner receives a set of loss (in Eqn. (3)) from the base learners and combine them to generate the output for round $t$. Let $p_t^i$ be the weight assigned to the $i$-th base learner at round $t$. The meta leaner outputs

$$\widehat{\phi}_t(\mathbf{x}_t) = \sum_i p_t^i \cdot \widehat{\phi}_t^i(\mathbf{x}_t). \tag{5}$$

The weight $p_t^i$ for each base-learner $f_t^i$ at round $t$ is updated by first calculating

$$p_t^i \propto \Phi(R_{t-1}^i + 1, C_{t-1}^i + 1) - \Phi(R_{t-1}^i - 1, C_{t-1}^i + 1), \tag{6}$$

where $\Phi(R, C) = \exp([R]_+^2 / 3C)$, $L_t = \sum_i p_t^i \cdot L_t^i$, and

$$R_t^i = R_{t-1}^i + (L_t - L_t^i), \quad C_t^i = C_{t-1}^i + |L_t - L_t^i|.$$

We set $R_t^i$ and $C_t^i$ to 0 when $t = 2^i$ for each restarted base learner.

---

**Algorithm 1** Adaptive Representation Alignment

---

1: **Initialization**: $\forall i \in [K], \phi_t^i = \phi_0$
2: **for** $t = 1$ **to** $T$ **do**
3:   **for** $i = 0$ **to** $K$ **do**
4:     **if** $2^i \bmod t == 0$ **then**
5:       set $\phi_t^i = \phi_0, R_t^i = 0, C_t^i = 0$
6:     **end if**
7:   **end for**
8:   update base learners by Eqn. (4) and update weight $\mathbf{p}_t \in \Delta_K$ according to Eqn. (6)
9:   combine base learners according to Eqn. (5)
10:   update classifier module according to Eqn. (7)
11: **end for**

---

After updating the representation learning model, we follow previous studies on TTA [32, 39, 25] by employing entropy minimization with a regularization term. This regularization ensures that the updated linear classification model remains close to the initial one, enhancing performance, which is defined as

$$\ell_{\text{entro}}^t(\mathbf{w}, \mathbf{x}) = -\sum_{y \in \mathcal{Y}} [\langle \mathbf{w}, \phi_t(\mathbf{x}) \rangle]_y \log([\langle \mathbf{w}, \phi_t(\mathbf{x}) \rangle]_y),$$

where $[\cdot]_y$ is taking the $y$-th entry of the vector $[\cdot]$. This regularization term encourages the model to generate confident predictions on unlabeled data by assigning higher probabilities to the most likely classes and lower probabilities to less likely ones. As a result, it helps prevent the model from becoming overly uncertain and making unreliable predictions. We then minimize the entropy of the predictions using the updated representation learning model with regularization, that is,

$$\mathbf{w}_t = \arg\min_{\mathbf{w}} \sum_{i=1}^{N_t} \ell_{\text{entro}}^t(\mathbf{w}, \mathbf{x}_i) + \|\mathbf{w} - \mathbf{w}_0\|_2. \tag{7}$$

We summarize the main procedures of the proposed algorithm in Algorithm 1.

**Remark 3** (Computational Efficiency). Since the $i$-th base learner is restarted every $2^i$-th round, within a time interval of size $T$, we only need to maintain at most $\log(T)$ base learners. For example, with $T = 100,000$, only 11 base learners are required. In the next section, we will demonstrate that this is sufficient to achieve the optimal dynamic regret for convex losses and models.

Note that the ensemble structure requires maintaining multiple base learners, to further enhance the computational efficiency, we follow the spirit of [42] to reduce the base learners' update complexity, which involves updating only the *affine parameters* of the normalization layers similar to those used in previous studies [21, 32, 28, 25]. The affine parameters typically comprise less than 1% of the total model parameters [32], making them particularly efficient to update.

### 3.4 Theoretical Analysis

In this part, we provide the theoretical analysis of our proposed Ada-ReAlign algorithm for convex losses and models. For convex representation learning models, such as input convex neural networks [1], and convex loss functions, we show that the proposed algorithm achieves a dynamic regret guarantee, using the optimal representation sequence $\{\phi_t^*\}_{t=1}^T$ as the comparator.

**Theorem 1.** *Assuming $\phi_t$ is convex, $L_t(\phi)$ is a convex with respect to $\phi$, and the input $\mathbf{x}_t$, the value of the loss function, and its gradient are all bounded. The Ada-ReAlign algorithm satisfies*

$$\mathbb{E}\left[\sum_{t=1}^T L_t(\phi_t) - \sum_{t=1}^T L_t(\phi_t^*)\right] \leq \mathcal{O}(T^{2/3} V_T^{1/3}),$$

*where function variation $V_T = \sum_{t=2}^T \sup_\phi |L_t(\phi) - L_{t-1}(\phi)|$. Detailed proofs are in **Appendix C.1**.*

Theorem 1 demonstrates that the average regret decreases at a rate of $\mathcal{O}(T^{-1/3})$. In this theorem, the function variation $V_T$ captures the cumulative change in the optimal representation function sequence,

serving as a measure of the underlying distribution shift in non-stationary environments and reflecting the inherent difficulty of the learning task. When the unlabeled test data stream is generated from a relatively stable environment, indicated by a small $V_T$, the accumulative loss decreases nearly at a rate of $\mathcal{O}(T^{2/3})$. We notice that directly optimizing Eqn. (3) in each round would result in $\mathcal{O}(T)$ regret, as a generalization error would accumulate in every round. Thus, Theorem 1 provides the theoretical foundation for the empirical success of the Ada-ReAlign algorithm in effectively adapting to unknown and continuous distribution shifts.

## 4 Experiments

We evaluate the proposed Ada-ReAlign algorithm on two large-scale benchmark datasets: CIFAR10C and ImageNetC. Our empirical studies aim to answer the following three questions: **Q1**: Does the Ada-ReAlign algorithm outperform competing methods? **Q2**: Are the mechanisms in the proposed algorithm effective in handling non-stationary data? **Q3**: Can the proposed method handle real-world data streams with unknown distribution shifts?

### 4.1 Experimental Setups

**Data Stream Generation.** The CIFAR10C and ImageNetC datasets provide both original clean data and corrupted data with varying types and levels of severity. We train the offline model on the clean data and use the corrupted data to generate unlabeled data streams, allowing us to simulate diverse distribution shifts within the data stream.

We assume a small batch of data is obtained at each round $t$, where $t \in [T] := \{1, ..., T\}$. In each round, this batch is generated from a fixed data distribution with a specific corruption type and severity level. By continuously varying the corruption types or severity levels, we simulate unlabeled data streams across different non-stationary environments. Let $N_t$ denote the number of data points in round $t$ and $M$ represent the duration for which the data distribution remains unchanged, spanning $M$ rounds between distribution shifts.

In our empirical studies, we simulate two common types of distribution shifts:

(1) *Gradual Shift*: To simulate an unlabeled data stream with a gradual shift, we keep the type of data corruption constant throughout the stream while varying the severity level every $M$ rounds. For example, in an unlabeled stream representing the "Snow" condition, the severity levels change sequentially as follows: $[1] * M \to [2] * M \to [3] * M \to ....$ Here, $[1] * M$ denotes $M$ consecutive rounds of data under the "Snow" condition with a severity level of 1.

(2) *Sequential Shift*: To simulate an unlabeled data stream with a sequential shift, we keep the corruption severity level constant throughout the stream while changing the type of data corruption every $M$ rounds. For instance, with a fixed severity level of 5, the corruption types evolve sequentially as follows: $[gaussian] * M \to [shot] * M \to [impulse] * M \to ....$ Here, $[shot] * M$ represents $M$ consecutive rounds of data with "shot" at severity level 5. In this paper, we set the severity level of the sequentially shifting data stream to 5.

**Contenders.** We compare the proposed algorithm with six competing methods. First, we use the performance of the initial *Non-adapt* model as a baseline. Next, we include methods that restart the model at each round and adapt based on the current round's data: *TENT-RE* [32], which minimizes entropy at each round, and *MEMO* [39], which enhances robustness through data augmentation. We also evaluate methods that leverage all the data, such as *TENT-CT*, which updates the model using the results from the previous round via the TENT [32] and *TTAC* [28], which focuses on aligning the representation with the initial model using the entire data stream. Additionally, we include two state-of-the-art TTA methods designed for non-stationary environments: *CoTTA* [34] and *SAR* [25], both of which incorporate a reset mechanism to mitigate long-term forgetting during adaptation.

**Implementation Details.** We conduct experiments using a deep neural network with a ResNet50 architecture from the torchvision library. The initial model is trained on the original CIFAR-10 and ImageNet datasets. For our proposed algorithm, we use SGD as the update rule, with a momentum of 0.9 and a learning rate of 0.0005. Following prior studies [21, 32, 28, 25], we adapt the *affine parameters* of the normalization layers in ResNet50 during the adaptation process. Further details are provided in **Appendix A.1**.

**Table 1:** The average classification error (in %) for the CIFAR10-to-CIFAR10C dataset under *Gradual Shift*. All results were averaged over 5 runs with different initializations. The number of data points per round was set to $N_t = 10$ with a duration of $M = 10$. The best results are highlighted in bold.

| Method | Gauss. | shot | impul. | defoc. | glass | motio. | zoom | snow | frost | fog | brigh. | contr. | elast. | pixel. | jpeg | Mean |
|---|---|---|---|---|---|---|---|---|---|---|---|---|---|---|---|---|
| Non-adapt | 36.6 | 30.8 | 30.6 | 13.8 | 42.0 | 18.3 | 17.2 | 20.3 | 22.7 | 15.1 | 11.1 | 16.3 | 17.1 | 23.5 | 22.7 | 21.5 |
|  | ±0.6 | ±0.4 | ±0.1 | ±1.7 | ±1.1 | ±0.8 | ±0.4 | ±0.3 | ±0.2 | ±0.4 | ±0.4 | ±0.1 | ±0.2 | ±0.1 | ±0.4 | ±0.8 |
| MEMO | 31.8 | 26.0 | 24.7 | 13.5 | 38.1 | 17.9 | 17.0 | 16.9 | 19.1 | 12.3 | 8.91 | 14.5 | 14.6 | 20.7 | 17.3 | 19.6 |
|  | ±0.4 | ±0.3 | ±0.2 | ±1.2 | ±1.0 | ±0.7 | ±0.6 | ±0.5 | ±0.3 | ±0.3 | ±0.5 | ±0.2 | ±0.3 | ±0.2 | ±0.6 | ±0.7 |
| TENT-RE | 32.4 | 25.1 | 24.1 | 14.2 | 37.6 | 18.1 | 17.2 | 15.4 | 19.7 | 11.9 | 9.20 | 13.7 | 14.2 | 20.4 | 18.1 | 18.5 |
|  | ±0.3 | ±0.3 | ±0.4 | ±1.5 | ±0.9 | ±1.1 | ±0.7 | ±0.6 | ±0.4 | ±0.2 | ±0.6 | ±0.3 | ±0.5 | ±0.3 | ±0.6 | ±0.9 |
| TENT-CT | 26.3 | 21.7 | 22.4 | 11.9 | 26.4 | 14.5 | 13.4 | 13.9 | 15.5 | 11.0 | 6.82 | 11.6 | 14.8 | 15.3 | 16.0 | 15.4 |
|  | ±0.3 | ±0.2 | ±0.3 | ±1.0 | ±0.8 | ±0.9 | ±0.5 | ±0.4 | ±0.3 | ±0.3 | ±0.5 | ±0.3 | ±0.4 | ±0.2 | ±0.5 | ±0.5 |
| TTAC | 24.6 | 22.0 | 22.2 | 10.8 | 25.7 | 12.7 | 10.1 | 12.9 | 14.4 | 10.7 | 5.52 | 10.3 | 14.2 | 13.8 | 15.1 | 14.3 |
|  | ±0.2 | ±0.4 | ±0.5 | ±0.9 | ±0.7 | ±0.9 | ±0.6 | ±0.5 | ±0.3 | ±0.4 | ±0.3 | ±0.1 | ±0.5 | ±0.3 | ±0.4 | ±0.5 |
| CoTTA | 22.1 | 20.6 | 23.0 | 9.80 | 25.3 | 10.7 | 8.08 | 12.2 | 12.6 | 10.2 | 5.90 | 8.20 | 14.3 | 12.5 | 15.3 | 14.1 |
|  | ±0.4 | ±0.3 | ±0.5 | ±1.1 | ±0.8 | ±0.8 | ±0.7 | ±0.4 | ±0.5 | ±0.3 | ±0.2 | ±0.3 | ±0.4 | ±0.2 | ±0.3 | ±0.6 |
| SAR | **20.6** | **19.3** | 21.9 | **8.00** | 24.1 | 9.13 | 5.96 | **10.4** | 10.7 | 9.21 | **4.12** | 6.92 | 13.2 | **11.3** | 13.4 | 12.6 |
|  | **±0.5** | **±0.3** | ±0.4 | **±1.0** | ±0.7 | ±0.6 | ±0.6 | **±0.5** | **±0.4** | ±0.3 | **±0.3** | ±0.4 | ±0.2 | **±0.3** | ±0.3 | ±0.5 |
| Ada-ReAlign | **20.6** | 19.8 | **21.1** | 8.19 | **23.4** | **8.72** | **5.30** | **10.4** | 11.1 | **9.04** | 4.51 | **6.29** | **12.6** | 12.8 | **12.7** | **11.8** |
|  | **±0.6** | ±0.5 | ±0.5 | ±1.2 | **±0.6** | **±0.8** | **±0.5** | **±0.4** | ±0.3 | ±0.1 | ±0.3 | ±0.5 | ±0.1 | ±0.2 | ±0.4 | **±0.7** |

**Table 2:** The average classification error (in %) for the CIFAR10-to-CIFAR10C dataset under *Sequential Shift*. All results were averaged over 5 runs with different initializations. The number of data points per round was set to $N_t = 10$ with a duration of $M = 10$. The best results are highlighted in bold.

| Method | Gauss. | shot | impul. | defoc. | glass | motio. | zoom | snow | frost | fog | brigh. | contr. | elast. | pixel. | jpeg | Mean |
|---|---|---|---|---|---|---|---|---|---|---|---|---|---|---|---|---|
| Non-adapt | 48.4 | 44.8 | 50.3 | 24.1 | 47.7 | 24.5 | 24.1 | 24.1 | 33.1 | 28.0 | 14.1 | 29.7 | 25.6 | 43.7 | 28.3 | 32.7 |
|  | ±0.8 | ±1.1 | ±0.2 | ±2.3 | ±1.8 | ±1.2 | ±0.4 | ±0.4 | ±0.6 | ±0.3 | ±0.5 | ±0.1 | ±0.3 | ±0.1 | ±0.7 | ±1.0 |
| MEMO | 43.5 | 39.8 | 43.3 | 26.4 | 44.4 | 25.1 | 25.0 | 20.9 | 28.3 | 22.8 | 11.9 | 28.3 | 21.1 | 42.8 | 21.7 | 30.4 |
|  | ±0.6 | ±1.0 | ±0.2 | ±0.9 | ±1.1 | ±1.1 | ±0.6 | ±0.3 | ±0.6 | ±0.5 | ±0.1 | ±0.5 | ±0.2 | ±0.6 | ±0.2 | ±1.1 |
| TENT-RE | 43.6 | 37.8 | 42.3 | 25.1 | 43.6 | 26.1 | 24.8 | 22.1 | 27.3 | 21.1 | 10.1 | 29.0 | 20.9 | 42.6 | 22.2 | 28.2 |
|  | ±2.0 | ±1.1 | ±0.3 | ±1.4 | ±1.4 | ±1.5 | ±0.0 | ±0.1 | ±1.0 | ±0.7 | ±0.4 | ±0.3 | ±0.2 | ±0.4 | ±0.5 | ±1.8 |
| TENT-CT | 38.6 | 34.4 | 42.4 | 28.2 | 44.9 | 30.3 | 27.9 | 32.9 | 32.4 | 32.4 | 26.0 | 34.1 | 39.5 | 34.6 | 38.7 | 34.2 |
|  | ±1.5 | ±0.9 | ±0.5 | ±1.1 | ±1.0 | ±1.2 | ±0.1 | ±0.2 | ±0.7 | ±0.5 | ±0.5 | ±0.6 | ±0.1 | ±0.4 | ±0.2 | ±1.4 |
| TTAC | 33.6 | 29.6 | 36.3 | 22.7 | 37.7 | 23.1 | 22.2 | 28.0 | 25.8 | 26.8 | 20.1 | 26.1 | 33.2 | 26.7 | 33.9 | 28.0 |
|  | ±1.0 | ±0.8 | ±0.3 | ±1.5 | ±1.6 | ±1.1 | ±0.4 | ±0.3 | ±0.9 | ±0.6 | ±0.4 | ±0.8 | ±0.4 | ±0.7 | ±0.6 | ±1.6 |
| CoTTA | 37.5 | 33.3 | 42.3 | 25.7 | 43.9 | 27.3 | 25.0 | 31.8 | 30.1 | 31.9 | 24.6 | 31.3 | 38.5 | 34.4 | 35.7 | 32.5 |
|  | ±0.8 | ±1.1 | ±0.8 | ±1.9 | ±1.2 | ±0.9 | ±0.6 | ±0.5 | ±0.1 | ±0.6 | ±0.7 | ±0.3 | ±0.6 | ±0.4 | ±0.6 | ±1.2 |
| SAR | 30.5 | 27.5 | 34.6 | 19.9 | 32.9 | 19.7 | 18.6 | 26.4 | 23.4 | 22.2 | 17.3 | 22.2 | 31.5 | 23.0 | 30.9 | 24.9 |
|  | ±1.1 | ±0.8 | ±0.7 | ±1.7 | ±1.0 | ±0.9 | ±0.5 | ±0.6 | ±0.9 | ±0.6 | ±0.4 | ±0.5 | ±0.4 | ±0.8 | ±0.3 | ±1.5 |
| Ada-ReAlign | **26.7** | **23.8** | **32.0** | **13.9** | **29.1** | **15.2** | **13.1** | **23.6** | **20.3** | **18.8** | **12.2** | **19.2** | **27.7** | **19.6** | **27.1** | **21.1** |
|  | **±1.4** | **±0.9** | **±0.6** | **±1.5** | **±1.3** | **±0.8** | **±0.3** | **±0.7** | **±0.8** | **±0.6** | **±0.2** | **±0.6** | **±0.3** | **±0.9** | **±0.2** | **±1.9** |

## 4.2 Performance Comparison

We present the comparison results for the CIFAR10-to-CIFAR10C datasets in Table 1 and Table 2. Results for the ImageNet-to-ImageNetC datasets are provided in **Appendix A.2**, where we observe similar trends. Our proposed Ada-ReAlign algorithm consistently achieves top performance under both gradual and sequential shifts, particularly in sequential shift scenarios, highlighting its adaptability in diverse non-stationary environments.

In Table 1, we report the average classification error rate, along with the standard deviation, for gradual severity shifts across 15 types of distribution shifts. These results are based on batches of $N_t = 10$ data points per round, with a distribution duration of $M = 10$ rounds, averaged over five runs with different initial models. Overall, our algorithm demonstrates competitive performance compared to other methods. We also observe performance improvements from TENT-CT and TTAC, which reuse all historical data, underscoring the importance of effectively leveraging past data. Meanwhile, Ada-ReAlign outperforms CoTTA and SAR, both of which incorporate reset mechanisms to enhance adaptability to severity shifts.

In Table 2, we evaluate the algorithm across the entire data stream and report its average accuracy for each type of data corruption. By effectively reusing historical data, our proposed Ada-ReAlign algorithm achieves a lower average classification error across all types of distribution shifts compared to the competitors. Notably, MEMO and TENT-RE, which reuse only the current batch of data, outperform TENT-CT, which utilizes all historical data. These findings align with our theoretical analysis, confirming that a fixed adaptation model is not well-suited for non-stationary environments.

Additionally, CoTTA, which stochastically resets parts of the model parameters to their initial values, also underperforms in non-stationary settings. This further highlights the superiority of our adaptive representation alignment algorithm. These results answer the question **Q1**.

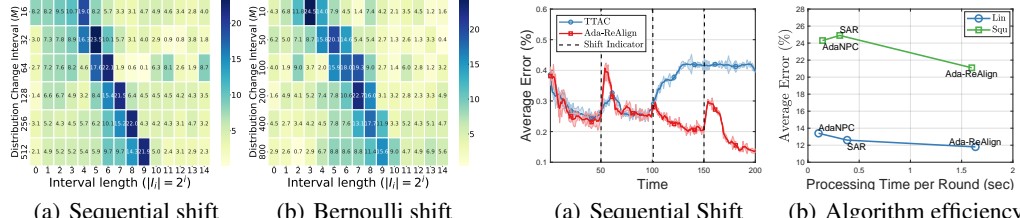

|  (a) Sequential shift | (b) Bernoulli shift |
| (a) Sequential Shift | (b) Algorithm efficiency |

**Figure 2:** Weight (%) heatmap of base learners in (a) Sequential shift with different intervals. (b) Bernoulli sequential shift with different intervals, where the length of interval is an expected value.

**Figure 3:** (a) Average error (%) and estimator loss curve with continuous sequential shift. (b) Average error (%) and processing time per round (sec) of three contenders with two kinds of distribution shifts.

### 4.3 Ablation Study

Next, we evaluate the adaptability of the Ada-ReAlign algorithm to changing distributions, along with its computational efficiency and the effectiveness of each component. Additional ablation studies on the impact of the number of data each round $N_t$ and duration $M$ are provided in **Appendix A.3**.

**Detailed Analysis of the Meta-Base Structure in Ada-ReAlign.** We now take a closer look at our adaptive representation alignment algorithm, which integrates a meta-base structure. Specifically, we conduct experiments to evaluate the algorithm's ability to handle sequential shifts, where the distribution of online data alternates between two distributions every $M$ rounds. To explore this, we vary the duration $M$ and report the weight assignment (expressed as percentages) for base learners with different interval lengths in Figure 2. For instance, when $M = 8$, the data distribution shifts every 8 rounds. The weights assigned to each base learner are averaged over its active period.

Our results show that the meta learner effectively assigns the highest weight to the base learner whose interval length aligns with the switch period $M$. This result shows that the right amount of historical data is reused, leading to strong performance in non-stationary environments.

In addition, in Figure 3 (a) compares the average error of the Ada-ReAlign algorithm with that of the TTAC algorithm over 200 iterations. During the experiment, the data distribution shifts three times, transitioning from Noise "Gaussian" to Blur "defocus", then to Weather "Snow", and finally to Digital "contrast". Each distribution is maintained for 50 iterations. TTAC is chosen for comparison since it also aims to align the model's representation with the initial one.

The results show that the Ada-ReAlign algorithm adapts quickly to new distributions as soon as a shift occurs. In contrast, the TTAC algorithm struggles, as it relies on all historical data, including samples from different distributions, which limits its adaptability.

**Efficiency and Performance Evaluation.** We conduct additional experiments to evaluate the algorithm's efficiency and performance gain. Specifically, we measure the processing time per round and the average accuracy, comparing the proposed algorithm with SAR (single model with forward and backward procedures) and AdaNPC (single forward procedure only) [40]. Since AdaNPC requires access to the labeled source data during adaptation, we exclude it from the main comparison.

As shown in Figure 3 (b), the Ada-ReAlign algorithm demonstrates superior performance in both gradual and sequential shifts on the CIFAR10-to-CIFAR10C dataset. In terms of computational time, Ada-ReAlign (with 11 base learners) is approximately five times slower than SAR. However, SAR involves solving a bi-level optimization problem during adaptation, which incurs additional computational overhead. AdaNPC is more efficient than both gradient-based algorithms, as it employs a KNN-based non-parametric classifier. These findings answer the question **Q2**.

**Component Analysis of the Proposed Algorithm.** We evaluate the impact of each component within the proposed algorithm in the sequential shift environment. The proposed algorithm consists of two elements: alignment of the representation divergence with the initial model and minimization of prediction entropy at each round. To investigate their individual contributions, we run the algorithm across the entire data stream using different loss configurations. Specifically, "Ada-ReAlign w/o DA" refers to the version where only prediction entropy is minimized at each step (without Distribution Alignment), while "Ada-ReAlign w/o EM" denotes the version where only the representation is aligned at each round (without Entropy Minimization).

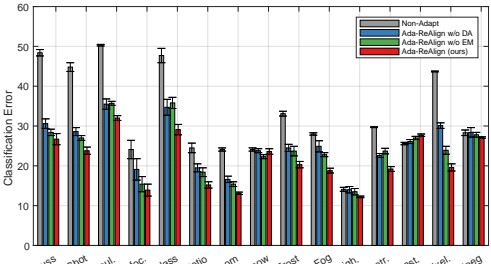
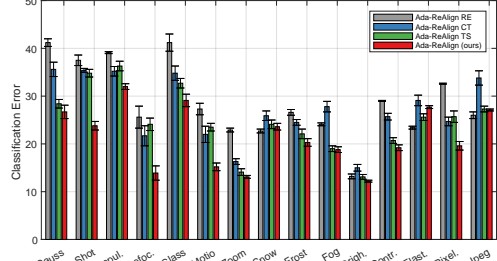

**Figure 4:** Average Classification Error (%) Comparison with Various Components.

**Figure 5:** Average Classification Error (%) Comparison with Different Restart Mechanisms.

We report the average accuracy for each type of data corruption. As shown in Figure 4, both representation alignment and entropy minimization play crucial roles in the performance of the Ada-ReAlign algorithm. Moreover, the results highlight that, in most cases of distribution shifts, representation alignment offers greater performance gains, underscoring the importance of adaptive representation alignment in non-stationary environments.

**Comparison with Restart Mechanisms.** We compare the proposed algorithm in a sequential shift environment with three different restart mechanisms: *Ada-RE*, *Ada-CT*, and *Ada-TS*. In *Ada-RE*, the model is reset to the initial offline model at the beginning of each round and undergoes "one-step" TTA using the surrogate loss defined in Eqn (3). This approach is similar to the MEMO and TENT-RE algorithms, which restart the model at each round.

The second method, *Ada-CT*, updates a single model continuously, where the adapted model from the previous round serves as the initial model for the next. The third approach, *Ada-TS*, also employs a single model but incorporates a restart mechanism based on a threshold for model entropy. Following the previous study [25], we restart the model whenever the entropy falls below a threshold of $0.4$.

The average accuracy of our proposed method and the three competing approaches is presented in Figure 5. The Ada-ReAlign algorithm consistently outperforms the competitors across nearly all datasets. These results highlight the critical role of the meta learner in the online ensemble structure, which enables the adaptive combination of base learners to resist the non-stationarity.

## 4.4 Real-World Evaluation on Wildlife Species Classification

We further evaluate the proposed algorithm on a real-world wildlife species classification task using the iWildCam dataset [3], where the distribution of images naturally varies with the time and location of capture. The earliest 10% of the data is used as labeled offline data to train the initial model, while the remaining data serves as the unlabeled data stream. We compare the performance of our method against competing approaches, and the results demonstrate that our algorithm achieves the best performance. These results answer question **Q3**.

**Table 3:** The Average Classification Error (%) for iWildCam dataset. All results were averaged over 5 runs with different initial models. We set number of data $N_t = 10$ at each round.

| Method | Non-adapt | TTAC | CoTTA | SAR | Ada-ReAlign |
|---|---|---|---|---|---|
| Classification Error (%) | $47.2 \pm 2.3$ | $27.3 \pm 1.9$ | $29.3 \pm 2.0$ | $31.5 \pm 2.4$ | $\mathbf{23.6 \pm 1.8}$ |

## 5 Conclusion

In this paper, we explored non-stationary representation learning for continual test-time adaptation. Beyond entropy minimization with regularization, we proposed adaptively aligning the unlabeled data stream, with its evolving distributions, to the source data representation by leveraging a sketch of the source data. To exploit this marginal information, we introduced a novel two-layer algorithm, Ada-ReAlign, designed to track and to approximate the optimal representation learning model at each round. Our theoretical analysis showed that the learned model is comparable to the optimal model sequence under convexity assumptions. Experiments on various benchmark datasets and a real-world application demonstrated the superiority of our approach over competing methods, confirming the effectiveness of the adaptive representation alignment mechanism.

## Acknowledgments

MS was supported by the Institute for AI and Beyond, UTokyo.

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

# Appendix

## A  Experiments

In this section, we supplement the omitted details in Section 4. We begin by presenting a comprehensive overview of the experimental setups, followed by showcasing the previously omitted empirical results on the ImageNet dataset. Furthermore, we conduct additional ablation studies to further investigate the proposed Ada-ReAlign algorithm.

### A.1  Experimental Setups

**Contenders.** We compare our proposed approach with six contenders, including a baseline method,

- *Non-adapt*: This method directly uses the offline model to make predictions on the unlabeled data stream without any adaptation.

One modified "one-round" TTA method is employed, where the model parameters are reset to the offline model at the start of each round and then undergo one-step adaptation:

- *MEMO* [39]: This method applies various data augmentations to each input and adapts all model parameters by minimizing the entropy of the average output distribution across these augmentations. At the beginning of each round, the model parameters are reset to those of the offline model.

Two continuous TTA methods are also evaluated, where the model is updated throughout the entire data stream without resets:

- *TENT* [32]: This method optimizes the channel-wise affine transformations within the normalization layers by minimizing the entropy of the model's predictions.
- *TTAC* [28]: This approach introduces a test-time anchored clustering method to improve the alignment of test-time features with the offline model's learned representation. It also employs pseudo-label filtering to enhance the effectiveness of the clustering process.

Additionally, we include two state-of-the-art continuous TTA methods specifically designed to address distribution shifts in non-stationary environments:

- *CoTTA* [34]: This algorithm continuously adapts by learning pseudo-labels from its own predictions. It leverages weight-averaged and augmentation-averaged predictions to produce more reliable pseudo-labels for the unlabeled data stream.
- *SAR* [25]: SAR is a sharpness-aware algorithm that minimizes entropy while ensuring robustness by filtering out noisy samples with large gradients. It encourages the model weights to converge to a flat minimum, improving resilience to noisy labeled data.

In summary, these contenders cover a wide range of continuous TTA approaches, from methods that rely on individual batches of data to those that incorporate the entire data stream. Our comparison also includes state-of-the-art algorithms specifically designed to handle distribution shifts in non-stationary environments.

**Algorithm Implementation Details of Contenders.** For the remaining contenders, we provide their detailed hyperparameter settings used in the experiments as follows.

- *MEMO* [39]: We adopted the default hyper-parameter settings used in MEMO. Specifically, we employed AugMix [13] for data augmentations, with an augmentation size of 32. The optimizer is SGD with a momentum of 0.9, a batch size of 32, and a learning rate of 0.0005.
- *TENT-RE* [32]: We adopted the default hyper-parameter settings used in TENT. The optimizer is SGD with a momentum of 0.9, a batch size of 32, and a learning rate of 0.0005. The trainable parameters are the affine parameters within the batch normalization layers. At the start of each round, the model parameters are reset to match those of the offline model.

**Table 4:** The average classification error (in %) for the ImageNet-to-ImageNetC dataset under *Gradual Shift*. All results were averaged over 5 runs with different initializations. The number of data points per round was set to $N_t = 10$ with a duration of $M = 10$. The best results are highlighted in bold.

| Method | Gauss. | shot | impul. | defoc. | glass | motio. | zoom | snow | frost | fog | brigh. | contr. | elast. | pixel. | jpeg | Mean |
|---|---|---|---|---|---|---|---|---|---|---|---|---|---|---|---|---|
| Non-adapt | 32.3 | 82.6 | 72.4 | 79.1 | 62.9 | 61.7 | 86.3 | 83.1 | 85.3 | 66.4 | 72.1 | 68.7 | 86.8 | 74.9 | 65.7 | 72.1 |
| | ±0.3 | ±0.4 | ±0.2 | ±0.6 | ±0.7 | ±0.5 | ±0.3 | ±0.2 | ±0.2 | ±0.3 | ±0.5 | ±0.6 | ±0.4 | ±0.3 | ±0.3 | ±0.5 |
| MEMO | 32.7 | 76.3 | 69.8 | 55.1 | 54.3 | 58.7 | 82.7 | 78.2 | 81.2 | 55.7 | 54.5 | 51.2 | 80.8 | 65.5 | 48.8 | 62.9 |
| | ±0.5 | ±0.8 | ±0.7 | ±0.9 | ±1.3 | ±0.5 | ±0.5 | ±0.6 | ±0.4 | ±0.2 | ±0.3 | ±0.3 | ±0.4 | ±0.5 | ±0.4 | ±0.6 |
| TENT-RE | 33.3 | 69.9 | 73.5 | 60.3 | 50.8 | 59.4 | 80.3 | 81.7 | 79.0 | 60.9 | 62.0 | 57.6 | 80.6 | 63.0 | 49.9 | 64.1 |
| | ±0.7 | ±1.0 | ±1.3 | ±1.2 | ±1.7 | ±1.4 | ±0.5 | ±0.5 | ±0.7 | ±0.4 | ±0.2 | ±0.6 | ±0.3 | ±0.5 | ±0.4 | ±0.6 |
| TENT-CT | 34.6 | 42.5 | 73.4 | 38.9 | 39.3 | 60.6 | 81.9 | 83.4 | 75.6 | 50.7 | 44.0 | 37.7 | 80.9 | 52.4 | 42.0 | 55.8 |
| | ±0.2 | ±0.6 | ±0.5 | ±0.8 | ±1.2 | ±1.6 | ±0.6 | ±0.5 | ±0.5 | ±0.4 | ±0.6 | ±0.7 | ±0.5 | ±0.6 | ±0.4 | ±0.8 |
| TTAC | 35.5 | 43.0 | 63.2 | 44.5 | 40.5 | 48.7 | 60.6 | 67.2 | 71.0 | 49.0 | 45.1 | 39.3 | 56.9 | 48.2 | 41.9 | 50.3 |
| | ±0.5 | ±0.3 | ±0.6 | ±1.7 | ±1.9 | ±0.7 | ±1.0 | ±0.8 | ±0.4 | ±0.4 | ±0.5 | ±0.3 | ±0.7 | ±0.4 | ±0.2 | ±0.8 |
| CoTTA | **32.6** | 40.4 | 64.1 | **37.3** | 36.4 | 46.5 | 63.4 | 69.3 | 64.2 | 46.5 | **40.8** | 38.5 | 61.7 | **44.1** | **39.1** | 48.3 |
| | ±**0.7** | ±**0.6** | ±0.4 | ±**2.0** | ±1.5 | ±0.9 | ±0.7 | ±0.3 | ±0.7 | ±0.3 | ±**0.2** | ±0.3 | ±0.3 | ±**0.3** | ±**0.6** | ±0.6 |
| SAR | 32.8 | 41.7 | 65.3 | 39.5 | **33.5** | 42.7 | 60.4 | 67.2 | 63.2 | 49.3 | 55.6 | 35.3 | 55.0 | 44.9 | 41.5 | 48.5 |
| | ±0.7 | ±0.5 | ±0.7 | ±1.6 | ±**1.3** | ±1.2 | ±1.2 | ±0.8 | ±0.6 | ±0.5 | ±0.5 | ±0.7 | ±0.3 | ±0.6 | ±0.7 | ±0.8 |
| Ada-ReAlign | 35.7 | 42.0 | **61.6** | 38.0 | 37.8 | **42.5** | **55.3** | **67.0** | **57.1** | **47.3** | 44.8 | **33.1** | **53.5** | 45.7 | 39.2 | **46.6** |
| | ±0.5 | ±0.6 | ±**0.6** | ±1.5 | ±1.6 | ±**1.8** | ±**0.4** | ±**0.6** | ±**0.5** | ±**0.3** | ±0.5 | ±**0.7** | ±**0.4** | ±0.3 | ±0.6 | ±**0.9** |

- *TENT-CT* [32]: We adopted the default hyper-parameter settings used in TENT. The optimizer is SGD with a momentum of 0.9, a batch size of 32, and a learning rate of 0.0005. The trainable parameters are the affine parameters within the batch normalization layers.

- *TTAC* [28]: We adopted the default hyper-parameter settings used in TTAC. The optimizer is SGD with a momentum of 0.9, a batch size of 32, and a learning rate of 0.0005. The trainable parameters are the affine parameters within the batch normalization layers.

- *CoTTA* [34]: We adopted the default hyper-parameter settings used in CoTTA. The optimizer is Adam with a learning rate of 0.001. We use the default data augmentation techniques in the CoTTA algorithm. We updated all trainable parameters while incorporating a restoration probability of $p = 0.01$ to mitigate the risk of catastrophic forgetting.

- *SAR* [25]: We adopted the default hyper-parameter settings used in SAR. The optimizer is SGD with a momentum of 0.9, a batch size of 32, and a learning rate of 0.0005. To clip the large gradients, we applied a threshold set to $0.4 \times \ln 1000$. Additionally, we set $\rho$ to 0.05 to encourage the optimization process to converge towards flat minima, following the default settings. For model recovery, we tracked the entropy loss values using a moving average factor of 0.9. The trainable parameters are the affine parameters within the batch normalization layers.

**Algorithm Implementation Details.** We adopt the standard ResNet-50 model from RobustBench [8] as the model structure. We freeze the top two linear layers and only update the affine parameters of the normalization layers within the remaining shallow layers of ResNet-50. The optimizer employed is SGD, with a momentum of 0.9 and a batch size of 32 for rounds where the number of data data exceeds 32. In cases where the number of data in a round is smaller than 32, we employ data augmentation techniques [39] to augment the data and ensure its number reaches a minimum of 32. The learning rate is set to 0.0005. We use the following compute resource configuration: 2 Xeon Gold 6242R with a base frequency of 3.1 GHz, 8 GeForce 3090 with 24GB VRAM, and a total of 768GB RAM. The operating system employed is Ubuntu 20.04.

### A.2    More Empirical Results on ImageNet-C for Section 4.2

Table 4 and Table 5 present the omitted results from Section 4.2. The results show that Ada-ReAlign outperforms the competing methods across nearly all tasks in the ImageNet-C dataset. These empirical findings confirm the effectiveness of our approach in adapting to non-stationary environments.

### A.3    Additional Ablation Studies

The number of data points in each round is a crucial measure of non-stationarity in the environment. As data accumulate continuously, and assuming that the data received in each round are generated from the same distribution, the number of data points per round serves as an indicator of the speed of distribution shifts. For a fixed duration $M$, a smaller number of data points per round suggests

**Table 5:** The average classification error (in %) for the ImageNet-to-ImageNetC dataset under *Sequential Shift*. All results were averaged over 5 runs with different initializations. The number of data points per round was set to $N_t = 10$ with a duration of $M = 10$. The best results are highlighted in bold.

| Method | Gauss. | shot | impul. | defoc. | glass | motio. | zoom | snow | frost | fog | brigh. | contr. | elast. | pixel. | jpeg | Mean |
|---|---|---|---|---|---|---|---|---|---|---|---|---|---|---|---|---|
| Non-adapt | 37.3 | 90.9 | 82.8 | 87.0 | 65.2 | 74.5 | 97.2 | 92.8 | 97.9 | 68.7 | 82.4 | 80.3 | 97.0 | 82.8 | 73.9 | 80.7 |
| | ±0.1 | ±0.1 | ±0.2 | ±0.4 | ±0.6 | ±0.5 | ±0.6 | ±0.2 | ±0.3 | ±0.3 | ±0.4 | ±0.2 | ±0.4 | ±0.2 | ±0.5 | ±0.4 |
| MEMO | 36.3 | 81.2 | 80.1 | 63.8 | 58.2 | 69.4 | 93.5 | 86.8 | 92.4 | 60.1 | 65.8 | 62.0 | 91.4 | 69.3 | 59.6 | 71.3 |
| | ±0.4 | ±0.9 | ±0.7 | ±0.5 | ±1.7 | ±0.6 | ±0.8 | ±1.2 | ±1.6 | ±0.6 | ±0.5 | ±0.4 | ±0.7 | ±0.5 | ±0.4 | ±0.9 |
| TENT-RE | 38.5 | 78.4 | 81.9 | 70.1 | 55.7 | 70.6 | 90.4 | 88.9 | 90.3 | 65.2 | 73.1 | 66.8 | 90.6 | 66.2 | 60.5 | 72.4 |
| | ±0.5 | ±1.5 | ±0.5 | ±1.2 | ±1.7 | ±1.2 | ±0.6 | ±0.6 | ±0.4 | ±1.5 | ±1.9 | ±1.1 | ±0.4 | ±0.8 | ±0.5 | ±1.2 |
| TENT-CT | 38.9 | 48.1 | 83.9 | 49.7 | 43.8 | 72.7 | 91.6 | 92.1 | 88.6 | 54.3 | 57.1 | 47.7 | 92.0 | 56.7 | 52.7 | 64.6 |
| | ±0.3 | ±1.9 | ±0.8 | ±1.1 | ±1.0 | ±1.2 | ±0.1 | ±0.2 | ±0.7 | ±0.5 | ±0.5 | ±0.6 | ±0.1 | ±0.4 | ±0.2 | ±1.4 |
| TTAC | 40.9 | 50.4 | 72.9 | 51.6 | 44.6 | 60 | 71.5 | 77.1 | 82.8 | 54.1 | 57.7 | 48.6 | 68.2 | 52.7 | 51.9 | 59.0 |
| | ±0.6 | ±1.8 | ±0.8 | ±0.7 | ±1.3 | ±0.9 | ±0.7 | ±0.5 | ±1.0 | ±0.4 | ±0.6 | ±0.4 | ±0.6 | ±0.4 | ±0.9 | ±1.0 |
| CoTTA | 36.6 | 46.7 | 73.4 | 46.9 | 38.9 | 57.6 | 73.7 | 78.5 | 76.9 | **49.3** | **52.8** | 48.9 | 72.8 | **48.4** | 49.6 | 56.7 |
| | ±0.4 | ±0.9 | ±0.6 | ±1.3 | ±1.8 | ±1.0 | ±1.1 | ±0.6 | ±0.7 | ±**0.6** | ±**0.4** | ±0.5 | ±0.4 | ±**0.4** | ±0.6 | ±0.9 |
| SAR | **36.2** | **46.3** | 74.0 | 47.2 | **37.2** | 54.0 | 68.8 | 76.6 | 74.4 | 51.6 | 65.9 | 45.3 | 65.7 | 48.8 | **49.1** | 56.0 |
| | ±**0.2** | ±**1.4** | ±1.3 | ±1.5 | ±**0.9** | ±1.3 | ±0.8 | ±0.6 | ±0.5 | ±0.3 | ±0.6 | ±0.5 | ±0.5 | ±0.6 | ±**0.8** | ±1.4 |
| Ada-ReAlign | 37.5 | 47.3 | **70.7** | **46.4** | 41.0 | **53.3** | **65.1** | **74.7** | **69.8** | 50.0 | 55.3 | **45.1** | **64.8** | 50.2 | **49.1** | **54.7** |
| | ±0.4 | ±1.3 | ±**1.3** | ±**1.1** | ±1.6 | ±**0.9** | ±**0.5** | ±**0.9** | ±**0.9** | ±0.5 | ±0.5 | ±**0.4** | ±**0.6** | ±0.4 | ±**0.7** | ±**1.2** |

**Table 6:** The Average Classification Error (%) for CIFAR10-to-CIFAR10C Dataset under `Sequential Shift`. All results were evaluated using the largest corruption severity level 5 and averaged over 5 runs with different initial models. We set different number of data $N_t$ at each round with duration $M = 10$.

| Method | Ada-ReAlign ($N_t$=1) | Ada-ReAlign ($N_t$=5) | Ada-ReAlign ($N_t$=10) | Ada-ReAlign ($N_t$=20) | Ada-ReAlign ($N_t$=50) |
|---|---|---|---|---|---|
| Classification Error (%) | 24.6 ± 2.7 | 21.1 ± 1.9 | 20.3 ± 1.5 | 19.5 ± 1.7 | 18.3 ± 2.5 |

**Table 7:** Performance Comparisons on CIFAR10-to-CIFAR10C continual test-time adaptation under different distribution shifts. All results were evaluated using the largest corruption severity level 5. 5 test runs were conducted with different initial models and the average classification error (%) as well as standard deviation are presented, with the best one emphasized in bold.

| | Method | Gauss. | shot | impul. | defoc. | glass | motio. | zoom | snow | frost | fog | brigh. | contr. | elast. | pixel. | jpeg | Mean |
|---|---|---|---|---|---|---|---|---|---|---|---|---|---|---|---|---|---|
| N=5 | SAR | 22.7 | 22.8 | 23.5 | **9.54** | 24.7 | 10.8 | **6.33** | 12.6 | **11.9** | 11.6 | **5.60** | 7.80 | 14.5 | **12.9** | 15.6 | 14.1 |
| | | ±0.8 | ±0.4 | ±0.3 | ±**0.7** | ±0.9 | ±0.5 | ±**0.6** | ±0.4 | ±**0.4** | ±0.5 | ±**0.6** | ±0.5 | ±0.6 | ±**0.2** | ±0.4 | ±0.7 |
| M=5 | Ours | **21.6** | **21.0** | **22.7** | 9.69 | **24.5** | **9.66** | 6.47 | **11.7** | **11.9** | **10.4** | 5.71 | **6.92** | **13.5** | 13.4 | **13.7** | **12.7** |
| | | ±**0.4** | ±**0.6** | ±**0.3** | ±0.8 | ±**0.5** | ±**0.7** | ±0.5 | ±**0.3** | ±**0.3** | ±**0.5** | ±0.5 | ±**0.5** | ±**0.4** | ±0.6 | ±**0.4** | ±**0.6** |
| N=5 | SAR | 21.6 | **20.2** | 22.6 | 8.79 | 24.8 | 9.88 | 6.84 | **10.9** | 11.6 | 10.8 | 5.79 | 7.73 | 13.5 | **12.3** | 14.6 | 13.1 |
| | | ±0.4 | ±**0.3** | ±0.3 | ±0.8 | ±0.4 | ±0.5 | ±0.6 | ±**0.7** | ±0.4 | ±0.6 | ±0.4 | ±0.3 | ±0.5 | ±**0.4** | ±0.3 | ±0.5 |
| M=10 | Ours | **21.0** | 20.5 | **21.8** | **8.56** | **24.1** | **9.33** | **5.98** | 11.4 | **11.5** | **9.80** | **5.37** | **7.04** | **12.9** | 12.8 | **13.4** | **12.5** |
| | | ±**0.6** | ±0.6 | ±**0.4** | ±**0.9** | ±**0.5** | ±**0.7** | ±**0.3** | ±0.6 | ±**0.3** | ±**0.4** | ±**0.5** | ±**0.3** | ±**0.5** | ±0.4 | ±**0.4** | ±**0.8** |
| N=10 | SAR | 21.4 | **19.5** | 22.3 | **8.35** | 24.3 | 9.61 | 6.53 | 11.4 | 11.9 | 9.70 | 4.63 | 6.84 | 13.8 | **11.5** | 13.4 | 13.2 |
| | | ±0.4 | ±**0.4** | ±0.4 | ±**0.9** | ±0.6 | ±0.7 | ±0.6 | ±0.3 | ±0.6 | ±0.4 | ±0.2 | ±0.5 | ±0.2 | ±**0.3** | ±0.6 | ±0.3 |
| M=5 | Ours | **20.9** | 19.7 | **21.8** | 8.81 | **23.9** | **8.56** | **5.69** | **10.6** | **11.5** | **9.42** | **4.37** | **6.51** | **12.34** | 12.4 | **12.9** | **11.8** |
| | | ±**0.5** | ±0.6 | ±**0.5** | ±1.4 | ±**0.9** | ±**0.6** | ±**0.7** | ±**0.7** | ±**0.5** | ±**0.2** | ±**0.2** | ±**0.5** | ±**0.4** | ±0.5 | ±**0.4** | ±**0.6** |
| N=10 | SAR | **20.6** | **19.3** | 21.9 | **8.00** | 24.1 | 9.13 | 5.96 | **10.4** | **10.7** | 9.21 | **4.12** | 6.92 | 13.2 | **11.3** | 13.4 | 12.6 |
| | | ±**0.5** | ±**0.3** | ±0.4 | ±**1.0** | ±0.7 | ±0.6 | ±0.6 | ±**0.5** | ±**0.4** | ±0.3 | ±**0.3** | ±0.4 | ±0.2 | ±**0.3** | ±0.3 | ±0.5 |
| M=10 | Ours | **20.6** | 19.8 | **21.1** | 8.19 | **23.4** | **8.72** | **5.30** | **10.4** | 11.1 | **9.04** | 4.51 | **6.29** | **12.6** | 12.8 | **12.7** | **11.8** |
| | | ±**0.6** | ±0.5 | ±**0.5** | ±1.2 | ±**0.6** | ±**0.8** | ±**0.5** | ±**0.4** | ±0.3 | ±**0.1** | ±0.3 | ±**0.5** | ±**0.1** | ±0.2 | ±**0.4** | ±**0.7** |

rapid distribution shifts, while a larger number indicates slower shifts, with a prolonged period during which data are generated from a consistent distribution.

Table 6 presents the performance of our Ada-ReAlign algorithm with varying numbers of data per round. Notably, even with limited data per round, the algorithm maintains relatively strong performance.

We further evaluate the algorithm's performance across different values of $M$ and $N$, where $M$ represents the length of time during which the data distribution remains constant, and $N$ is the number of data points available at each time step. Throughout the experiments, we maintain a batch size of 32. When $N \leq 32$, the batch size is adjusted to $N$. A smaller $M$ indicates more frequent distribution shifts within the unlabeled data stream.

From Tables 7 and 8, we observe that the Ada-ReAlign algorithm consistently outperforms its closest competitor, the SAR algorithm, across various values of $M$ and $N$, especially in scenarios involving sequential shifts. Both SAR and Ada-ReAlign experience performance degradation as distribution shifts become more frequent and fewer data points are available per time step. However, Ada-ReAlign exhibits more modest degradation compared to SAR, demonstrating greater robustness in the face of rapid distribution shifts and limited data availability at each step.

**Table 8:** Performance Comparisons on CIFAR10-to-CIFAR10C continual test-time adaptation under different distribution shifts. All results were evaluated using the largest corruption severity level 5. 5 test runs were conducted with different initial models and the average classification error (%) as well as standard deviation are presented, with the best one emphasized in bold.

| | Method | Gauss. | shot | impul. | defoc. | glass | motio. | zoom | snow | frost | fog | brigh. | contr. | elast. | pixel. | jpeg | Mean |
|---|---|---|---|---|---|---|---|---|---|---|---|---|---|---|---|---|---|
| N=5 | SAR | 35.5 | 31.2 | 37.2 | 22.4 | 35.1 | 22.3 | 20.5 | **27.3** | 25.7 | 24.7 | 18.9 | 24.5 | 33.9 | 25.2 | 32.4 | 28.5 |
| | | ±1.3 | ±1.3 | ±0.8 | ±1.5 | ±1.2 | ±1.2 | ±0.8 | **±1.1** | ±0.9 | ±0.8 | ±0.4 | ±0.5 | ±0.3 | ±0.9 | ±0.4 | ±2.4 |
| M=5 | Ours | **30.8** | **27.3** | **36.5** | **16.8** | **32.6** | **17.9** | **17.9** | 27.5 | **23.5** | **21.7** | **14.6** | **22.8** | **31.5** | **23.5** | **31.4** | **24.8** |
| | | **±1.7** | **±1.9** | **±1.5** | **±1.8** | **±1.4** | **±1.6** | **±0.6** | **±0.9** | **±0.8** | **±0.8** | **±0.4** | **±0.6** | **±0.6** | **±1.3** | **±0.4** | **±2.2** |
| N=5 | SAR | 35.4 | 30.7 | 37.0 | 21.9 | 34.4 | 21.5 | 19.9 | 27.2 | 25.3 | 23.8 | 18.5 | 23.7 | 33.2 | 24.6 | 31.7 | 27.9 |
| | | ±0.9 | ±0.8 | ±1.2 | ±1.4 | ±1.2 | ±0.8 | ±0.6 | ±0.6 | ±0.8 | ±1.1 | ±0.6 | ±0.7 | ±0.5 | ±1.2 | ±0.6 | ±1.7 |
| M=10 | Ours | **28.0** | **25.2** | **34.6** | **15.0** | **30.4** | **16.7** | **15.3** | **25.2** | **22.1** | **19.9** | **13.5** | **20.4** | **28.8** | **20.4** | **28.6** | **22.5** |
| | | **±1.2** | **±1.2** | **±0.8** | **±1.7x** | **±1.1** | **±0.5** | **±0.5** | **±0.6** | **±0.8** | **±0.7** | **±0.6** | **±0.5** | **±0.4** | **±0.8** | **±0.3** | **±1.8** |
| N=10 | SAR | 32.3 | 28.6 | 35.1 | 20.6 | 33.5 | 20.2 | 19.3 | 26.9 | 24.8 | 23.1 | 17.8 | 22.9 | 32.6 | 23.8 | 31.3 | 27.1 |
| | | ±1.5 | ±0.7 | ±0.5 | ±1.5 | ±1.2 | ±0.6 | ±0.4 | ±0.6 | ±0.8 | ±1.1 | ±0.6 | ±0.5 | ±0.3 | ±1.0 | ±0.6 | ±1.7 |
| M=5 | Ours | **27.8** | **24.3** | **32.5** | **14.6** | **29.8** | **15.7** | **13.5** | **24.0** | **21.1** | **19.5** | **13.1** | **19.8** | **28.2** | **19.9** | **27.8** | **22.8** |
| | | **±1.6** | **±1.1** | **±0.4** | **±1.3** | **±1.8** | **±0.7** | **±0.4** | **±0.7** | **±0.6** | **±0.8** | **±0.4** | **±0.5** | **±0.5** | **±0.9** | **±0.8** | **±2.0** |
| N=10 | SAR | 30.5 | 27.5 | 34.6 | 19.9 | 32.9 | 19.7 | 18.6 | 26.4 | 23.4 | 22.2 | 17.3 | 22.2 | 31.5 | 23.0 | 30.9 | 24.9 |
| | | ±1.1 | ±0.8 | ±0.7 | ±1.7 | ±1.0 | ±0.9 | ±0.5 | ±0.6 | ±0.9 | ±0.6 | ±0.4 | ±0.5 | ±0.4 | ±0.8 | ±0.3 | ±1.5 |
| M=10 | Ours | **26.7** | **23.8** | **32.0** | **13.9** | **29.1** | **15.2** | **13.1** | **23.6** | **20.3** | **18.8** | **12.2** | **19.2** | **27.7** | **19.6** | **27.1** | **21.1** |
| | | **±1.4** | **±0.9** | **±0.6** | **±1.5** | **±1.3** | **±0.8** | **±0.3** | **±0.7** | **±0.8** | **±0.6** | **±0.2** | **±0.6** | **±0.3** | **±0.9** | **±0.2** | **±1.9** |

# B   Related Work

In this section, we introduce the relevant literature on test-time adaptation and learning algorithms designed for non-stationary environments.

**TTA in Stationary Environment.** TTA considers the problem in which the source labeled data are no longer accessible during the adaptation phase. In this task, the learner relies solely on the source model and seeks to adapt it to a fixed test dataset with a distinct distribution. TTA methods can be broadly categorized into two groups based on whether they require a specific training process.

Test-Time Training [30] is the first group, involves optimizing the initial model with a combination of supervised and self-supervised losses, followed by self-supervised learning during test time to update the model. Common self-supervised losses include rotation prediction [30], contrastive self-supervised learning [23], and others.

The second group of methods does not require a specific training process and can be directly applied to any neural network. These methods include adapting batch normalization statistics [26, 15, 22], entropy minimization [32, 25, 39], pseudo-labeling [34, 5], and more.

However, these approaches are designed for adaptation to a fixed distribution and are not well-suited for non-stationary environments, where the data distribution evolves over time.

**Continual TTA in Non-stationary Environments.** In recent years, TTA in non-stationary environments has gained significant attention. As streaming data evolve over time, the underlying data distribution naturally shifts. A straightforward approach involves resetting the model's parameters to those of the initial model after each mini-batch adaptation, as seen in methods like MEMO [39], episodic TENT [32], and DDA [10]. However, these one-step adaptation methods often fall short due to high variance from the limited data available per round and their inability to leverage accumulated historical data for improved performance.

To address these challenges, CoTTA [34] was introduced, employing robust pseudo-labels generated through a weighted average of historical models. CoTTA also preserves the initial model's parameters by stochastically replacing the model's parameters with those of the initial model at each round. Similarly, SAR [25] estimates reliable entropy and resets the model to its initial state when the entropy exceeds a predefined threshold. AdaNPC [40] tackles non-stationary shifts by constructing a memory to store historical distributions. Additionally, continual test-time adaptation techniques have been extended to large language models [9].

While these approaches have demonstrated empirical success in various real-world tasks, the challenge of effective representation alignment in non-stationary environments remains underexplored.

**Adapting to Non-stationary Environments with Offline Labeled Data.** Online convex optimization [11] provides a powerful paradigm to handle sequential prediction problems. Over the decades, a variety of online learning algorithms have been proposed to handle the changing environments by optimizing the dynamic regret measure [46, 4, 43, 35] under different kinds of feedback information and

non-stationarity measures of environments. When data is weakly labeled, handling non-stationarity becomes extremely challenging as it is hard to estimate the loss at each round.

This line of research then consider certain kinds of distribution shift and adaptive learning with weighted source labeled data or test unlabeled data with pseudo labels to adapt to continuous distribution change. Previous work tackled the challenge of non-stationarity in the context of online label shift, where only the class prior is changing. In such scenarios, an unbiased loss estimator is constructed to estimate the loss at each round, enabling the use of dynamic regret minimization to handle non-stationary environments [2]. Another work investigated the problem of continuous covariate shift where only the input distribution changes [41]. They reframed this problem as an online density ratio estimation task and proposed a generic reduction of the density ratio estimation problem to dynamic regret optimization. However, these methods may not always be feasible to access offline labeled data for online adaptation.

In contrast to label shift or covariate shift, gradual domain adaptation consider the gradual shift of underlying distribution. An early work in this area is continuous manifold adaptation [14], which considers adaptation to evolving domains. More recently, self-training has shown promising results by pseudo labeling the unlabeled data stream and adaptively re-train the model based on the pseudo labeled data [19]. Theoretical analysis demonstrated that when the optimal classifier only undergoes slight shifts between consecutive batches, self-training can provide a well-generalized classifier throughout the entire data stream [36, 33]. However, these approaches assume that the data stream satisfies this assumption, and may not handle outliers that deviate from it. To handle such outliers, several approaches employ active learning [6, 44] to alleviate their negative impact. Nevertheless, these methods are designed for specific types of data distribution change problems.

## C  Theoretical Analysis

In this section, we present the theoretical analysis of the guarantees for our proposed algorithm. The analysis is based on the convex optimization framework, assuming both the model and the loss function are convex. While our method does not strictly fit within the convex optimization framework, the analysis offers valuable insights that inform the design of an effective algorithm. Furthermore, the theoretically guided approach demonstrates strong empirical performance [44, 41].

We first introduce the following assumptions for our analysis.

**Assumption 1.** The gradients of all functions are bounded by $G$, i.e.,

$$\max_{\phi \in \Phi} \|\nabla \ell_t(\phi)\|_2 \leq G, \forall t \in [T].$$

**Assumption 2.** The domain $\Phi$ contains the origin $0$, and its diameter is bounded by $D$, i.e.,

$$\max_{\phi, \phi' \in \Phi} \|\nabla \phi - \phi'\|_2 \leq D.$$

**Assumption 3.** The value of each function belongs to $[0, 1]$, i.e.

$$0 \leq \ell_t(\phi) \leq 1, \forall \phi \in \Phi, t \in [T].$$

As long as the loss functions are bounded, they can always be scaled and restricted to $[0, 1]$.

### C.1  Proof of Theorem 1

In this part, we provide the detailed proof for Theorem 1.

*Proof.* As we use the information each round the update the model, this operation implicitily is an optimistic mirror descent where the optimism is the gradient of the empirical loss at that time. We firstly show that these two framework are equal, so that we can analyze our proposed algorithm with the optimistic mirror descent framework. The optimistic mirror descent works as follows, for $t = 1, ..., T$,

$$\phi_t = \Pi[\widehat{\phi}_t - \eta M_t]$$
$$\widehat{\phi}_{t+1} = \Pi[\widehat{\phi}_t - \eta f_t(\phi_t)]$$

where $f_t(\cdot)$ is the loss function each time. Therefore, we prove that our method is equal to optimistic mirror descent by choosing $M_t = \nabla \widehat{L}_t(\widehat{\phi}_t)$. In the following, we use the optimistic mirror descent framework to analyze it.

Now we begin to prove the dynamic regret of the representation learning model. We firstly prove the static regret for the base model. Under the Assumption 1, 2 and 3, according to Lemma 2, we have the following inequality for any comparator $\mathbf{u}$ in any interval $I$,

$$\mathbb{E}[\sum_{t \in I} L_t(\phi_t) - \sum_{t \in I} L_t(\mathbf{u})] \leq \mathbb{E}[\sum_{t \in I} \langle \nabla \widehat{L}(\phi_t), \phi_t - \mathbf{u} \rangle + \sum_{t \in I} 2(LD + G)\|\phi_t - \widehat{\phi}_t\|_2]. \quad (8)$$

For the first term in Eqn. (8), according to we have the following inequality

$$\sum_{t \in I} \langle \nabla \widehat{L}(\phi_t), \phi_t - \mathbf{u} \rangle$$
$$\leq \sum_{t \in I} \langle \nabla \widehat{L}(\phi_t) - M_t, \phi_t - \widehat{\phi}_{t+1} \rangle + \langle M_t, \phi_t - \widehat{\phi}_{t+1} \rangle + \langle \nabla \widehat{L}(\phi_t), \widehat{\phi}_{t+1} - \mathbf{u} \rangle$$
$$\leq \frac{G\eta}{2}|I| + \frac{D}{2\eta}$$

where the second inequality holds due to Theorem 1 in [43].

For the second term in Eqn. (8), considering our update rule $\phi_t = \Pi(\widehat{\phi}_t - \eta M_t)$, where $M_t = \nabla \widehat{L}_t(\widehat{\phi}_t)$, according to the Lemma 1, we have

$$\|\phi_t - \widehat{\phi}_t\|_2 \leq \eta\|M_t\|_2 \leq \eta G.$$

Therefore, we have

$$\mathbb{E}[\sum_{t \in I} L_t(\phi_t) - \sum_{t \in I} L_t(\mathbf{u})] \leq \mathbb{E}[\frac{3G\eta}{2}|I| + \frac{D}{2\eta}]. \quad (9)$$

By choosing $\eta = \mathcal{O}(\sqrt{|I|})$, we have

$$\mathbb{E}\left[\sum_{t \in I} L_t(\phi_t) - \sum_{t \in I} L_t(\mathbf{u})\right] \leq \mathcal{O}(\sqrt{|I|}).$$

Secondly, as we use the AdaNormalHedge as the meta learner, according to Lemma 3, we can obtain the following strongly adaptive regret

$$\mathbb{E}[\sum_{t \in I} L_t(\phi_t) - \sum_{t \in I} L_t(\phi_I^*)] \leq \mathcal{O}(\sqrt{|I|}).$$

Now we use the guarantee of adaptive regret to obtain the dynamic regret. Let $V_T = \sum_{t=2}^{T} \sup_\phi |L_t(\phi) - L_{t-1}(\phi)|$ is the functional variation, we have

$$\mathbb{E}\left[\sum_{t=1}^{T} L_t(\phi_t) - \sum_{t=1}^{T} L_t(\phi_t^*)\right]$$
$$\leq \mathbb{E}\left[\sum_{m=1}^{M} \sum_{t \in I_m} L_t(\phi_t) - \sum_{m=1}^{M} \sum_{t \in I_m} L_t(\phi_{\mathcal{I}_m}^*) + \sum_{m=1}^{M} \sum_{t \in I_m} L_t(\phi_{\mathcal{I}_m}^*) - \sum_{t=1}^{T} L_t(\phi_t^*)\right] \quad (10)$$

where in the first inequality we introduce a split of the whole time horizon, which can be arbitrary and we only use it in the analysis. In the second inequality, the fisrt term is the adaptive regret used in Lemma 3. For the second term can, we can follow the reasoning in [4] and subsequently simplified analysis in [43, Theorem 7] to show that

$$\sum_{m=1}^{M} \sum_{t \in I_m} L_t(\phi_t^*) - \sum_{t=1}^{T} L_t(\phi_t^*) = \sum_{m=1}^{M} \sum_{t \in \mathcal{I}_m} \left(L_t(\phi_{\mathcal{I}_m}^*) - L_t(\phi_t^*)\right)$$

$$
\leq \sum_{m=1}^{M} \sum_{t \in \mathcal{I}_m} \left( L_t(\phi_{s_m}^*) - L_t(\phi_t^*) \right)
$$

$$
= \sum_{m=1}^{M} \sum_{t \in \mathcal{I}_m} \left( L_t(\phi_{s_m}^*) - L_{s_m}(\phi_{s_m}^*) + L_{s_m}(\phi_{s_m}^*) - L_t(\phi_t^*) \right)
$$

$$
\leq \sum_{m=1}^{M} \sum_{t \in \mathcal{I}_m} \left( L_t(\phi_{s_m}^*) - L_{s_m}(\phi_{s_m}^*) + L_{s_m}(\phi_t^*) - L_t(\phi_t^*) \right)
$$

$$
\leq 2 \frac{T}{M} \sum_{m=1}^{M} \sum_{t \in \mathcal{I}_m} \sup_{\phi} |L_t(\phi) - L_{t-1}(\phi)|
$$

$$
= 2 \frac{T}{M} \sum_{t=2}^{T} \sup_{\phi} |L_t(\phi) - L_{t-1}(\phi)|
$$

$$
\triangleq 2B \frac{T}{M} V_T^R, \tag{11}
$$

where $s_m = (m-1)T/M + 1$ is the first time step at interval $\mathcal{I}_m$. In the above, the first inequality is due to the optimality of $\phi_{\mathcal{I}_m}^*$ over the interval $\mathcal{I}_m$. The second inequality holds since $\phi_{s_m}^* \in \arg\min_{\phi} R_{s_m}(\phi)$.

Combining Eqn. (10) and Eqn. (11), we can obtain

$$
\mathbb{E}\left[ \sum_{t=1}^{T} L_t(\phi_t) - \sum_{t=1}^{T} L_t(\phi_t^*) \right]
$$

$$
\leq \mathbb{E}\left[ \sum_{m=1}^{M} \sqrt{\frac{T}{M}} + \frac{T}{M} V_T \right]
$$

$$
\leq \mathcal{O}(T^{2/3} V_T^{1/3})
$$

The third inequality is due to the AM-GM inequality. Now we complete the proof. $\qquad \square$

### C.2 Technical Lemmas

This section provides several useful technical lemmas used in the proof. The first three lemmas are the concentration on each time-stamp while the final one is a general inequality.

**Lemma 1** (Stability lemma [7, Proposition 7]). *Consider the following two updates: (i)* $\mathbf{x}_* = \arg\min_{\mathbf{x} \in \mathcal{X}} \langle \mathbf{a}, \mathbf{x} \rangle + B_\psi(\mathbf{x}, \mathbf{c})$, *and (ii)* $\mathbf{x}_*' = \arg\min_{\mathbf{x} \in \mathcal{X}} \langle \mathbf{a}', \mathbf{x} \rangle + B_\psi(\mathbf{x}, \mathbf{c})$. *When the regularizer* $\psi : \mathcal{X} \to \mathbb{R}$ *is a 1-strongly convex function with respect to the norm* $\| \cdot \|$, *we have* $\|\mathbf{x}_* - \mathbf{x}_*'\| \leq \|(\nabla \psi(\mathbf{c}) - \mathbf{a}) - (\nabla \psi(\mathbf{c}) - \mathbf{a}')\|_* = \|\mathbf{a} - \mathbf{a}'\|_*$.

**Lemma 2** (Lemma 2 in [44]). *Under the Assumption 1, 2 and 3, we have the following inequality for* $\mathbf{u}_t, t = 1, ...T$,

$$
\mathbb{E}[L_t(\phi_t) - L_t(\phi_I^*)] \leq \mathbb{E}[\langle \nabla \widehat{L}(\phi_t), \phi_t - \mathbf{u}_t \rangle] + \mathbb{E}[2(LD + G)\|\phi_t - \widehat{\phi}_t\|_2].
$$

**Lemma 3** (Theorem 3 in [38]). *Under the Assumption 1, 2 and 3, running AdaNormalHedge as the meta learner, we have the following inequality for any interval I that*

$$
\mathbb{E}\left[ \sum_{t \in I} L_t(\phi_t) - \sum_{t \in I} L_t(\phi_I^*) \right] \leq \mathcal{O}(\sqrt{|I|}).
$$

## Impact Statements

This research investigates a general machine learning problem of test-time adaptation, where we consider the continuous distribution shift in the unlabeled data stream. The consequences of system failure and bias in the data are not applicable.

