# OpenReview forum: "Test-time Adaptation in Non-stationary Environments via Adaptive Representation Alignment"
_NeurIPS.cc/2024/Conference — NeurIPS 2024 poster_

### Official Review · Reviewer_rMcB · 2024-07-10

**Soundness:** 3
**Presentation:** 2
**Contribution:** 3
**Rating:** 6
**Confidence:** 3

**Summary:**

This paper proposes a test-time adaptation (TTA) method focusing on adapting to a non-stationary environment.
The proposed method DART aims at minimizing the cumulative loss over time steps, which is equivalent to minimize the distribution gap between the source and current features.
To capture unknown changing periods of the environment, DART employs multiple base learners that are reset at different periods.
DART incorporates the outputs of the base learners with dynamically learned weights to make final predictions.
Experimental results show that DART can follow continual environmental changes on image corruption datasets.

**Strengths:**

- S1: Addressing continual adaptation of non-stational environments is practical.
- S2: Providing the theoretical insights is beneficial.
- S3: It is interesting that minimizing the distribution gap results in minimizing the regret.

**Weaknesses:**

- W1: Equations should be presented accurately.
  - The argument of $\hat{L}_t$ in Eq. (4) seems to be $\hat{L}\_t(\phi\_t,\phi\_0)$.
  - Softmax seems to be missing in the formulation of $\ell\_\text{entro}^t$.
  - Ada-ReAlign minimizes the norms of the feature statistics between the source and current distributions, but Fig. 1 says that it minimizes KL divergence.
- W2: Providing more detailed explanations of theoretical analyses in Sec. 3.4 would be beneficial. For example, where do $\mathcal{O}(T^{-1/3})$ in L221 and $\mathcal{O}(T)$ in L227 come from?
- W3: Ada-ReAlign requires to keep $\log T$ models, which is more than the baseline methods. Comparing memory consumption would be fair.
- W4: Can we estimate the distribution gap in Eq. (3) accurately? When the batch size is smaller than the number of feature dimensions, the covariance matrix degenerates, i.e., the covariances cannot be estimated as we can take arbitrary subspaces that include all samples in the batch. Even if numerical stability is improved by adding a small constant to the diagonals of the covariance matrix (L153), this problem still remains unless the batch size is larger than the feature dimensions. It would be beneficial to evaluate the distribution gap with other metrics, e.g., optimal transport or MMD, at each round.

**Questions:**

- Q1: How do we determine the number of the base learners when $T$ is unknown?
- Q2: Can we incorporate the idea of the meta-learner and base learners with other TTA methods?

**Limitations:**

Discussed in Appendix.

---

> ### Author Rebuttal · Authors · 2024-08-06
>
> Thank you for your detailed review and thoughtful feedback. We will revise the manuscript accordingly. Specific questions are answered below.
>
> ---
>
> **Question in W2:** Providing more detailed explanations of theoretical analyses in Sec. 3.4 would be beneficial. For example, where do $\mathcal{O}(T^{-1/3})$ in L221 and $\mathcal{O}(T)$ in L227 come from?
>
> **Answer for W2:** Thanks for the valuable suggestions. We will provide a brief explanation below and include a detailed discussion in the revised paper.
>
> As shown in Theorem 1, the *cumulative regret* over the entire time horizon of length $T$ is of the order $\mathcal{O}(T^{2/3}V_T^{1/3})$, where $V_T$ is a algorithm-independent variable measuring the degree of the non-stationarity of the environment. Dividing by $T$, the *average regret* is then of the order $\mathcal{O}(T^{-1/3})$ in L221.
>
> If we optimize Eqn. (3) directly each round, using only data from that round and not incorporating historical data, we will incur $\mathcal{O}(T)$ cumulative regret. This is because the algorithm suffers a generalization error independent of $T$ in each round, because it uses empirical data only in that round for optimization, and this error accumulates over the entire time horizon, resulting in a cumulative regret of $\mathcal{O}(T)$ in L227. Unlike using only the current round of data, our method explores the historical data, providing an opportunity to make the cumulative regret bound scale to the order of $\mathcal{o}(T)$.
>
> ---
>
> **Question in W4:** Can we estimate the distribution gap in Eq. (3) accurately? When the batch size is smaller than the number of feature dimensions, the covariance matrix degenerates...
>
> **Answer for W4:** Thank you for the insightful comments. Besides the metric we used, several effective methods can also be incorporated, such as optimal transport and MMD mentioned. Determining which one is superior remains an open question, particularly in continuous adaptation cases where the data is limited at each timestamp, posing a common challenge for all estimation methods. We will discuss them and propose alternative methods below.
>
> In this paper, inspired by previous work showing satisfactory performance [1,2], we use a Gaussian distribution to approximate the representation distribution. The covariance matrix may degenerate when the batch size is significantly smaller than the number of feature dimensions. In such cases, we can use a pre-trained feature extractor to compress the original features to a size smaller than the batch size, or use data augmentation techniques [3] to increase the batch size for more accurate discrepancy estimation.
>
> MMD is a good alternative measure, but it may require careful design as it may be less accurate in measuring the distribution gap if the chosen kernel is inappropriate by definition. Since the use of other metrics is also feasible, we leave a detailed study of these for future work.
>
> We will revise the paper accordingly and include this discussion in the main paper.
>
> ---
>
> **Q1:** How do we determine the number of the base learners when $T$ is unknown?
>
> **A1:** Thank you for pointing this out. The number of base learners can dynamically increase when $T$ is unknown. There are also mechanisms to handle cases where the number of base learners is constrained by resource limitations. Below, we propose alternative methods to enable the handling of an unknown $T$.
>
> We can use the doubling trick [4] to handle the case when $T$ is unkown. Before running the algorithm, we initially set a value for $T$, e.g., $T=100,000$, then execute the algorithm. When the timestamp reaches $T$, we restart all the base learners and the meta learner, double the value of $T$, i.e., set $T = 2T$, introducing a new base learner, and rerun the algorithm. Theoretically, the bound will suffer an additional logarithmic term, changing from $\mathcal{O}(T^{2/3}V_T^{1/3})$ to $\mathcal{O}(T^{2/3}V_T^{1/3}\log T)$. Since $\log T$ is a relatively small term, its effect is minor.
>
> When $T$ is very large and the number of base learners is fixed by resource limitations, we can eliminate some frequently restarted base learners with restart intervals of $1$ or $2$, which are designed to ensure a dynamic regret bound in worst-case scenarios. In real-world tasks, however, we can drop these frequently restarted base learners since the distribution usually does not change very frequently.
>
> We will revise the draft accordingly to add the above discussion in the main paper.
>
> ---
>
> **Q2:** Can we incorporate the idea of the meta-learner and base learners with other TTA methods?
>
> **A2:** The two-layer structure is a general idea to handle unknown and continous distribution changes and can be incorporated into other TTA methods encountering such continous changes during deployment. For instance, given a black-box TTA algorithm $\mathcal{A}$ and a measurement $M$, we can initialize a set of base learners using algorithm $\mathcal{A}$ with varying restart intervals (such as geometric covering), and then run them simultaneously during testing. At each time step, we use $M$ to evaluate each base learner and adjust their corresponding weights accordingly. Intuitively, regardless of whether the distribution changes rapidly or slowly, the base learner that explores an appropriate period of time will achieve better performance. This design could allow the black-box algorithm to automatically adapt to unknown distribution changes.
>
> ---
>
>
> **Reference**
>
> [1] Su, Yongyi, Xun Xu, and Kui Jia. Revisiting realistic test-time training: Sequential inference and adaptation by anchored clustering. In: NeurIPS, 2022.
>
> [2] Zellinger, Werner, et al. Central Moment Discrepancy (CMD) for Domain-Invariant Representation Learning. In: ICLR, 2022.
>
> [3] Zhang, Hongyi, et al. Mixup: Beyond Empirical Risk Minimization. In: ICLR, 2018.
>
> [4] Hazan, Elad. Introduction to online convex optimization. Foundations and Trends in Optimization, 2016.

---

> > ### Comment · Reviewer_rMcB · 2024-08-11
> >
> > Thank you for your rebuttal and further explanation.
> > I would like to increase the score.

---

> > > ### Author Response · Authors · 2024-08-11
> > >
> > > Dear Reviewer rMcB,
> > >
> > > Thank you again for your valuable feedback, which has definitely improved our paper. We will revise the manuscript accordingly to incorporate our discussion into the updated version.
> > >
> > > We are grateful for the time and effort you put into reviewing our work. Thank you very much!
> > >
> > > Best regards,
> > >
> > > Authors

---

### Official Review · Reviewer_azfj · 2024-07-11

**Soundness:** 3
**Presentation:** 3
**Contribution:** 3
**Rating:** 7
**Confidence:** 4

**Summary:**

This paper proposes a novel algorithm called Ada-ReAlign for sequentially adapting a model to non-stationary environments. The proposed algorithm uses a set of base learners, each equipped with different learning window sizes, with a meta learner that combines their outputs. This online ensemble allows an adaptive projection of the unlabeled data, which exhibits changing distributions, onto the source distribution. Subsequently, entropy minimization is applied to the unlabeled data, facilitating the refinement of representations according to the updated distributions. This paper provides theoretical analysis and empirical experiments to justify and validate the effectiveness of the proposed approach.

**Strengths:**

+ This paper is well organized and easy to follow, all the problem formulation and details of the proposed algorithm are well discussed, the experimental setup is clear to me and the result is convincing.

+ The transformation of continuous test-time adaptation to dynamic regret minimization for representation alignment is novel to me. Based on this transformation, the proposed approach is versatile and capable of handling non-stationary environments without explicitly defining or identifying distribution shift boundaries.

+ The proposed approach is theoretically sound. The dynamic exploration of different sliding window sizes of the data stream is novel and intriguing to me. The algorithm is sound with nice theoretical guarantees.

+ Empirical results show superiority over competing algorithms on two simulated benchmark datasets and a wildlife species classification task of the iWildCam dataset. Many ablation studies are performed and the results seem convincing.

**Weaknesses:**

- The proposed algorithm requires the sketch of the source domain feature embeddings during the adaptation, which may limit the use of the proposed algorithm in practice.

- Although the proposed approach uses only a small number of base learners, $n$ base learners means $n$ times slower than a traditional base learner algorithm, which increases the computational burden.

**Questions:**

Should the number of base learners used depend on the downstream problem? How is the number of base models determined and the size of the sliding window chosen? Large windows adapt to gradual changes over time, while small windows adapt to drastic changes. It seems that the window size does not depend on the specific problem.

The meta-learner seems rather complicated. Why this particular design?

In the proposed algorithm, the number of base learners seems to be fixed. What is the performance when the number of experts is limited to a fixed number, less than the required number?

**Limitations:**

Please see Weaknesses.

---

> ### Author Rebuttal · Authors · 2024-08-06
>
> Thank you for your detailed review and thoughtful feedback. We will revise the manuscript accordingly. Specific questions are answered below.
>
> ---
>
> **Q1:** Should the number of base learners used depend on the downstream problem? How is the number of base models determined and the size of the sliding window chosen? Large windows adapt to gradual changes over time, while small windows adapt to drastic changes. It seems that the window size does not depend on the specific problem.
>
> **A1:** Indeed, the number of base learners and the size of the sliding window depend on the properties of the downstream problem, such as the type and interval of the distribution shifts. If we have prior knowledge of these properties, we can adjust the number of base learners and the window size accordingly. For instance, if the distribution shift is piece-wise, we can use a single base learner and restart it whenever the shift occurs.
>
> Unfortunately, in real-world applications, the property of the downstream problem is usually **unknown and constantly changing**. Since the prior knowledge is unknown, we propose to use a set of base learners and a meta learner to allow the learning system to automatically adapt to unknown changes. We consider nearly all possible kinds of shifts and design the number of base learners and the corresponding window size based solely on the length of the entire time horizon. The proposed method can **automatically** adapt to the downstream problem without prior knowledge, since the meta learners can dynamically identify the most appropriate base learner.
>
> This design is based on the dense geometric covering method [1] from the online non-stationary learning literature. The dense geometric covering generates a set of intervals with varying lengths that divide the entire sequence of size $T$, and the algorithm will create a base learner for each interval. Since our meta-learner can automatically track the best-performing base learner, it can adapt to different distribution changes by tracking the corresponding base learner. Intuively, when the distribution change slowly and the base learner in a rather long interval performs well, the proposed method will sign higher weights for this base learner. An illustration of dense geometric covering is provided in the PDF.
>
> We will revise the draft accordingly to add the above discussion in the main paper.
>
>
> ---
>
> **Q2:** The meta-learner seems rather complicated. Why this particular design?
>
> **A2:** The meta learner is designed to handle unknown distribution changes and the unknown optimal time-varying competitor. We can use Hedge [2] to simplify the meta learner, but AdaNormalHedge offers better properties both practically and theoretically. We use AdaNormalHedge because it is a parameter-free expert-tracking algorithm. It has two main advantages:
> 1. *Parameter-Free Nature*: AdaNormalHedge does not require prior information. Unlike other meta algorithms such as Hedge, which achieve optimal regret only when the learning rate is finely tuned based on the competitor, AdaNormalHedge as a meta algorithm is parameter-free. This means that we do not need prior knowledge of the optimal competitor or the number of experts in advance.
> 2. *More Adatpive Bound*: AdaNormalHedge provides a more adaptive bound that allows us to obtain a problem-dependent regret bound. This means that if the problem is relatively simple, with the optimal base learner experiencing small losses over the entire time horizon, the proposed algorithm can achieve a tighter regret bound.
>
>
> ---
>
> **Q3:** In the proposed algorithm, the number of base learners seems to be fixed. What is the performance when the number of experts is limited to a fixed number, less than the required number?
>
> **A3:** In this case, we can eliminate some frequently restarted base learners with restart intervals of $1$ or $2$. These base learners are designed to ensure a dynamic regret bound in worst-case scenarios. In real-world tasks, however, we can drop these frequently restarted base learners because the distribution usually does not change so adversely.
>
> ---
>
> **Reference**
>
> [1] Zhang, Lijun, Shiyin Lu, and Tianbao Yang. Minimizing dynamic regret and adaptive regret simultaneously. In AISTATS, 2020.
>
> [2] Yoav Freund and Robert E. Schapire. Adaptive game playing using multiplicative weights. Games and Economic Behavior, 1999.

---

> ### Comment · Reviewer_azfj · 2024-08-13
>
> Thank you for your rebuttal. I will increase the score.

---

> > ### Author Response · Authors · 2024-08-13
> >
> > Dear Reviewer azfj,
> >
> > Thank you again for your valuable feedback, which has definitely improved our paper. We will revise the manuscript accordingly to incorporate our discussion into the updated version.
> >
> > We are grateful for the time and effort you put into reviewing our work. Thank you very much!
> >
> > Best regards,
> >
> > Authors

---

### Official Review · Reviewer_73Jf · 2024-07-11

**Soundness:** 3
**Presentation:** 3
**Contribution:** 3
**Rating:** 7
**Confidence:** 5

**Summary:**

The paper investigates test-time adaptation in a non-stationary environment where unlabeled data batches arrive sequentially with changing data distributions. The authors propose non-stationary representation learning to project the changing distribution back to the original source data distribution and update the classifier with entropy minimization based on the projected representation. Theoretical analysis shows that the proposed method has solid dynamic regret guarantees. Experiments show that the method of representation adaptation is effective in mitigating the distribution shift.

**Strengths:**

1. The continuous test-time adaptation problem studied in the paper is a popular and crucial area of research for real-world applications of domain adaptation.

2. The proposed non-stationary representation learning is novel and offers non-trivial technical contributions. The authors adapt techniques from online ensemble learning to test-time adaptation, addressing the challenges posed by unknown distribution changes and the problems of high bias and high variance due to the limited amount of unlabeled data available in each round in the data streams.

3. The effectiveness of the method is validated through experiments on both benchmark datasets and a real-world application. In addition, its computational efficiency is well discussed.

**Weaknesses:**

1. The paper lacks a sufficient review of related work, particularly in the online learning literature. For instance, the method uses dynamic regret as a performance measure and uses techniques from online ensembles. However, readers unfamiliar with this area or these techniques may find it difficult to understand the underlying motivation.

2. The theoretical guarantees focus primarily on convex models and losses, making it hard to adapt them to deep learning models.

3. In the sequential shift, the cyclicality period is fixed by $M$. The performance under conditions of sudden changes or non-cyclic changes is not well studied. Ablation studies are mainly conducted on benchmark datasets, leaving the performance in real-world applications unclear.

**Questions:**

1. How does Theorem 1 apply to deep models, considering that the theory was originally formulated for linear models?

2. In the sequential shift, the cyclicality period is fixed by $M$. How does the performance change when the cyclicality period varies?

3. How stable is the performance of the proposed method on the iWildCam dataset with different hyperparameters? are there any ablation studies?

**Limitations:**

Yes

---

> ### Author Rebuttal · Authors · 2024-08-06
>
> Thank you for your detailed review and thoughtful feedback. We will revise the manuscript accordingly. Specific questions are answered below.
>
> ---
>
> **Q1:** How does Theorem 1 apply to deep models, considering that the theory was originally formulated for linear models?
>
> **A1:** Theorem 1 shows that AdaReAlign minimizes dynamic regret under convexity assumptions, allowing the learned model sequence to approximate the optimal one. This allows AdaReAlign to effectively adapt to unknown and continuously changing distributions. While these theoretical results depend on convexity assumptions, it may be possible to extend them to deep models by exploiting results on neural tangent kernels. However, this extension is beyond the primary scope of our paper. In addition, many studies show that the analysis for convex models can guide the algorithm design, and the convexity assumption does not limit the practical performance of the algorithm with non-convex models [1,2,3].
>
> ---
>
> **Q2:** In the sequential shift, the cyclicality period is fixed by $M$. How does the performance change when the cyclicality period varies?
>
> **A2:** Thank you for your suggestions. We have added additional experiments in which the cyclicality period is sampled randomly from a fixed mean, and tested the proposed algorithm against other contenders. The experimental results are reported in Table 1 in the supplementary PDF. The results demonstrate that the proposed method is robust and also achieves better performance in this setting.
>
> We will revise the draft accordingly to include the above results in the paper.
>
> ---
>
> **Q3:** How stable is the performance of the proposed method on the iWildCam dataset with different hyper-parameters? are there any ablation studies?
>
> **A3:** Thank you for your suggestions. We evaluated the hyper-parameters on the iWildCam dataset to test the performance of our proposed algorithm. The results are reported in Table 2 in the supplementary PDF. The findings demonstrate that the proposed method is rather stable.
>
> We will revise the draft accordingly to include the above results in the paper.
>
> ---
>
> **Reference**
>
> [1] Kingma, Diederik P., and Jimmy Ba. Adam: A method for stochastic optimization. In: ICLR, 2015
>
> [2] Sun, Y, et al. Test-time training with self-supervision for generalization under distribution shifts. In: ICML, 2021
>
> [3] Bai, Y, et al. Adapting to online label shift with provable guarantees. In: NeurIPS, 2022

---

> > ### Comment · Reviewer_73Jf · 2024-08-10
> >
> > After reviewing the responses and the revisions made, I am satisfied with how they have addressed the concerns. I am willing to accept the manuscript and recommend a score adjustment to reflect the improvements.

---

> > > ### Author Response · Authors · 2024-08-11
> > >
> > > Dear Reviewer 73Jf,
> > >
> > > Thank you again for your valuable feedback, which has definitely improved our paper. We will revise the manuscript to include the new supplementary results according to your suggestions.
> > >
> > > We are grateful for the time and effort you put into reviewing our work. Thank you very much!
> > >
> > > Best regards,
> > >
> > > Authors

---

### Author Rebuttal · Authors · 2024-08-06

We are very grateful to all reviewers for your valuable feedback, which has definitely helped in improving our paper. We respond to the questions raised respectively and report all the tables and figures in the supplementary PDF.

---

### Decision · Program_Chairs · 2024-09-25

**Decision:**

Accept (poster)

**Comment:**

This paper proposes a novel algorithm called Ada-ReAlign for sequentially adapting a model to non-stationary environments. The proposed algorithm uses a set of base learners, each equipped with different learning window sizes, with a meta-learner that combines their outputs. The reviewers all believe that the proposed method could be a valuable contribution to the community.  I thus recommend accepting this paper.